EMBO
Molecular Medicine

# β-RA reduces DMQ/CoQ ratio and rescues the encephalopathic phenotype in *Coq9R239X* mice

Agustín Hidalgo-Gutiérrez[1,2], Eliana Barriocanal-Casado[1,2], Mohammed Bakkali[3], M Elena Díaz-Casado[1,2], Laura Sánchez-Maldonado[1,2], Miguel Romero[4], Ramy K Sayed[2,5], Cornelia Prehn[6], Germaine Escames[1,2,7], Juan Duarte[4], Darío Acuña-Castroviejo[1,2,7] & Luis C López[1,2,7,*] (iD)

## Abstract

Coenzyme Q (CoQ) deficiency has been associated with primary defects in the CoQ biosynthetic pathway or to secondary events. In some cases, the exogenous CoQ supplementation has limited efficacy. In the *Coq9R239X* mouse model with fatal mitochondrial encephalopathy due to CoQ deficiency, we have tested the therapeutic potential of β-resorcylic acid (β-RA), a structural analog of the CoQ precursor 4-hydroxybenzoic acid and the anti-inflammatory salicylic acid. β-RA noticeably rescued the phenotypic, morphological, and histopathological signs of the encephalopathy, leading to a significant increase in the survival. Those effects were due to the decrease of the levels of demethoxyubiquinone-9 ($DMQ_9$) and the increase of mitochondrial bioenergetics in peripheral tissues. However, neither CoQ biosynthesis nor mitochondrial function changed in the brain after the therapy, suggesting that some endocrine interactions may induce the reduction of the astrogliosis, spongiosis, and the secondary down-regulation of astrocytes-related neuroinflammatory genes. Because the therapeutic outcomes of β-RA administration were superior to those after $CoQ_{10}$ supplementation, its use in the clinic should be considered in CoQ deficiencies.

**Keywords** 2,4-dihydroxybenzoic acid; astrogliosis; mitochondrial encephalopathy; Q synthome; spongiosis
**Subject Categories** Genetics, Gene Therapy & Genetic Disease; Pharmacology & Drug Discovery

## Introduction

Mitochondria are the primary site of cellular energy production and can also have a broad range of other vital functions for the cells. Dysfunction of mitochondria is observed in common age-related disorders, including neurodegenerative diseases, metabolic syndrome, cardiomyopathies, and sarcopenia (Nunnari & Suomalainen, 2012). Mitochondrial dysfunction is likewise the primary characteristic of monogenic disorders that affect one of the genes that encode the estimated 1,000 proteins identified in mitochondria (Pagliarini *et al*, 2008). Among these, mutations in genes encoding essential proteins for the oxidative phosphorylation (OXPHOS) system account for the most common group of inborn errors of metabolism. The therapeutic options for these disorders are mostly limited to palliative care and remain woefully inadequate (DiMauro *et al*, 2013; Wang *et al*, 2016).

Primary deficiency of coenzyme Q10 ($CoQ_{10}$), due to $CoQ_{10}$ biosynthetic defects, is a mitochondrial syndrome that affects the OXPHOS system, and it has heterogeneous clinical presentations and variable response to therapy with exogenous $CoQ_{10}$ (DiMauro *et al*, 2007; Emmanuele *et al*, 2012). To understand the disease mechanisms, we previously generated and characterized the first mouse model of mitochondrial encephalopathy due to CoQ deficiency. The mouse model presents two point mutations, leading to a premature stop codon in *Coq9* (*Coq9R239X*; Garcia-Corzo *et al*, 2013). These mutations are homologues to the mutations identified in a patient (*COQ9R244X*; Duncan *et al*, 2009). The resulting dysfunctional COQ9 protein causes a profound reduction in the levels of COQ7 protein and disruption of the multiprotein complex for CoQ biosynthesis (Complex Q), resulting in a severe CoQ deficiency with accumulation of demethoxyubiquinone (DMQ = 2-polyprenyl-6-methoxy-3-methyl-1,4-benzoquinone), brain mitochondrial dysfunction, oxidative damage, disruption of sulfide

1 Departamento de Fisiología, Facultad de Medicina, Universidad de Granada, Granada, Spain
2 Instituto de Biotecnología, Centro de Investigación Biomédica, Universidad de Granada, Granada, Spain
3 Departamento de Genética, Facultad de Ciencias, Universidad de Granada, Granada, Spain
4 Departamento de Farmacología, Facultad de Farmacia, Universidad de Granada, Granada, Spain
5 Department of Anatomy and Embryology, Faculty of Veterinary Medicine, Sohag University, Sohag, Egypt
6 Institute of Experimental Genetics, Genome Analysis Center, Helmholtz Zentrum München, Neuherberg, Germany
7 Centro de Investigación Biomédica en Red de Fragilidad y Envejecimiento Saludable (CIBERFES), Granada, Spain
*Corresponding author. Tel: +34 958241000 ext. 20197; E-mail: luisca@ugr.es

metabolism, reactive astrogliosis, spongiform degeneration, hypotension, and premature death (Garcia-Corzo et al, 2013; Luna-Sanchez et al, 2017). These pathological features are similar to those described in patients with mutations in COQ7 and COQ9, who presented a severe early-onset multisystemic disease dominated by encephalopathy, the phenotype associated with CoQ deficiency with the most limited response to oral CoQ$_{10}$ supplementation (Lopez et al, 2006; Duncan et al, 2009; Emmanuele et al, 2012; Danhauser et al, 2016; Smith et al, 2018).

Consistent with the poor therapeutic effects of CoQ supplementations in patients, oral supplementation with ubiquinone-10, the oxidized form of CoQ$_{10}$, was ineffective in the Coq9$^{R239X}$ mouse model due to its poor absorption and bioavailability, which is more accused in the CNS because of the presence of the blood brain barrier (Bentinger et al, 2003; Bhagavan & Chopra, 2007; Lopez et al, 2010; Garcia-Corzo et al, 2014). The supplementation with ubiquinol-10, the reduced form of ubiquinone-10, provided better therapeutic outcomes because it induced slight improvements in mitochondrial bioenergetics in the brain and reduction of the oxidative stress and astrogliosis (Garcia-Corzo et al, 2014).

An alternative strategy to the exogenous CoQ$_{10}$ supplementation is to increase the endogenous CoQ biosynthesis. The biosynthesis of CoQ$_{10}$ occurs in a complex biosynthetic pathway in which the 4-hydroxybenzoic acid (4-HB) is the initial substrate. This precursor of the benzoquinone ring of CoQ is first prenylated, and then, a total of seven reactions (one decarboxylation, three hydroxylation, and three methylation) produce the fully substituted benzoquinone ring of CoQ (Fig EV1; Kawamukai, 2016). One of the hydroxylation steps is catalyzed by COQ7, a hydroxylase that needs another protein, COQ9, for its stability and normal function (Garcia-Corzo et al, 2013; Lohman et al, 2014). The hydroxyl group incorporated by COQ7 into the benzoquinone ring is already present in the molecule β-resorcylic acid (β-RA), also named 2,4-hydroxybenzoic acid (2,4-diHB). Therefore, β-RA is a 4-HB analog that can be theoretically used in the CoQ biosynthesis pathway in order to bypass a defect from the hydroxylation step catalyzed by COQ7 (Fig EV1). This strategy was partially successful in vitro in COQ7 null yeasts, as well as in mouse and human fibroblasts with mutations in COQ7 or COQ9 (Xie et al, 2012; Freyer et al, 2015; Luna-Sanchez et al,

2015). Additionally, β-RA is a structural analogue of salicylic acid (2-hydroxybenzoic acid), an anti-inflammatory molecule that may be potentially valuable to reduce the neuroinflammation (Lan et al, 2011; Gomez-Guzman et al, 2014), a factor that has been recently postulated as an essential pathomechanism and a therapeutic target in mitochondrial encephalopathies (Ignatenko et al, 2018), leukoencephalopathy, and neurodegenerative diseases (Eto et al, 2002; Liddelow & Barres, 2015, 2017). Also salicylic derivate, known as salicylates, may act as mTOR inhibitors (Cameron et al, 2016), and this action may be therapeutic in mitochondrial diseases (Johnson et al, 2013).

Based on the properties of β-RA, we have evaluated its therapeutic potential in a mouse model of mitochondrial encephalopathy due to CoQ deficiency. Our data show novel therapeutic mechanisms and represent one of the few cases of successful therapies for mitochondrial encephalopathies that could easily be implemented into the clinic.

## Results

### The treatment with β-RA rescues the phenotype of Coq9$^{R239X}$ mice

We previously demonstrated that Coq9$^{R239X}$ mice have reduced size and body weight, and die between 3 and 7 months of age (Garcia-Corzo et al, 2013). Moreover, we proved that ubiquinol-10 treatment induces biochemical and histopathological improvements in the brain of Coq9$^{R239X}$ mice (Garcia-Corzo et al, 2014). These improvements led to a significant increase in the survival of Coq9$^{R239X}$ mice treated with ubiquinol-10, compared to the Coq9$^{R239X}$ mice. While the Coq9$^{R239X}$ mice reached a maximum age of 7 months, with a median survival of 5 months of age, the Coq9$^{R239X}$ mice treated with ubiquinol-10 reached a maximum age of 17 months, with a median survival of 13 months of age (Fig 1A). Interestingly, the therapeutic effect of oral β-RA supplementation resulted in an even greater increase in the survival of Coq9$^{R239X}$ mice, reaching a maximum lifespan of 25 months of age, with a median survival of 22 months of age (Fig 1A). Therefore, the extension in the lifespan achieved by β-RA in Coq9$^{R239X}$ mice reached values close to the lifespan in wild-type mice (Fig 1A).

Figure 1. Survival and phenotypic characterization of Coq9$^{R239X}$ mice after β-RA treatment.

A    Survival curve of the Coq9$^{+/+}$ mice, Coq9$^{R239X}$ mice, and Coq9$^{R239X}$ mice after β-RA treatment and Coq9$^{R239X}$ after ubiquinol-10 treatment. The treatments started at 1 month of age [Log-rank (Mantel–Cox) test or Gehan–Breslow–Wilcoxon test; Coq9$^{+/+}$, n = 9; Coq9$^{R239X}$, Coq9$^{R239X}$ after β-RA treatment, and Coq9$^{R239X}$ after ubiquinol-10 treatment, n = 15 for each group].

B    Survival curve of the Coq9$^{R239X}$ mice and Coq9$^{R239X}$ mice after β-RA treatment started at 3 months of age (Coq9$^{R239X}$, n = 15; Coq9$^{R239X}$ after β-RA treatment, n = 10).

C, D    Body weight of males (C) and females (D) Coq9$^{+/+}$ mice, Coq9$^{R239X}$ mice, and Coq9$^{R239X}$ mice after β-RA treatment [(C): Coq9$^{+/+}$, n = from 31 to 4; Coq9$^{R239X}$, n = from 13 to 13; Coq9$^{R239X}$ after β-RA treatment, n = from 31 to 7. (D): Coq9$^{+/+}$, n = from 22 to 3; Coq9$^{R239X}$, n = from 13 to 12; Coq9$^{R239X}$ after β-RA treatment, n = from 24 to 3].

E, F    Rotarod test of male (E) and female (F) Coq9$^{+/+}$ mice, Coq9$^{R239X}$ mice, and Coq9$^{R239X}$ mice after β-RA treatment [(E): Coq9$^{+/+}$, n = from 11 to 8; Coq9$^{R239X}$, n = from 10 to 10; Coq9$^{R239X}$ after β-RA treatment, n = from 11 to 9. (F): Coq9$^{+/+}$, n = from 5 to 5; Coq9$^{R239X}$, n = from 4 to 4; Coq9$^{R239X}$ after β-RA treatment, n = from 11 to 6].

G    Blood pressure of Coq9$^{+/+}$ mice, Coq9$^{R239X}$ mice, and Coq9$^{R239X}$ mice after β-RA treatment. Data from male and female mice are represented together (Coq9$^{+/+}$, n = 6; Coq9$^{R239X}$, n = 6; Coq9$^{R239X}$ after β-RA treatment, n = 7). The whiskers down and up of the boxes indicate the minimum and maximum value, respectively. Horizontal lines inside the boxes mark the median.

H    Comparative image of a Coq9$^{R239X}$ mouse and a Coq9$^{R239X}$ mouse after β-RA treatment at 4 months of age.

Data information: Data are expressed as mean ± SD. *P < 0.05; **P < 0.01; ***P < 0.001; Coq9$^{R239X}$ or Coq9$^{R239X}$ after β-RA treatment versus Coq9$^{+/+}$. $^{+}$P < 0.05; $^{++}$P < 0.01; Coq9$^{R239X}$ versus Coq9$^{R239X}$ after β-RA treatment (one-way ANOVA with a Tukey's post hoc test or t-test; see Appendix Table S2 for exact P-values).

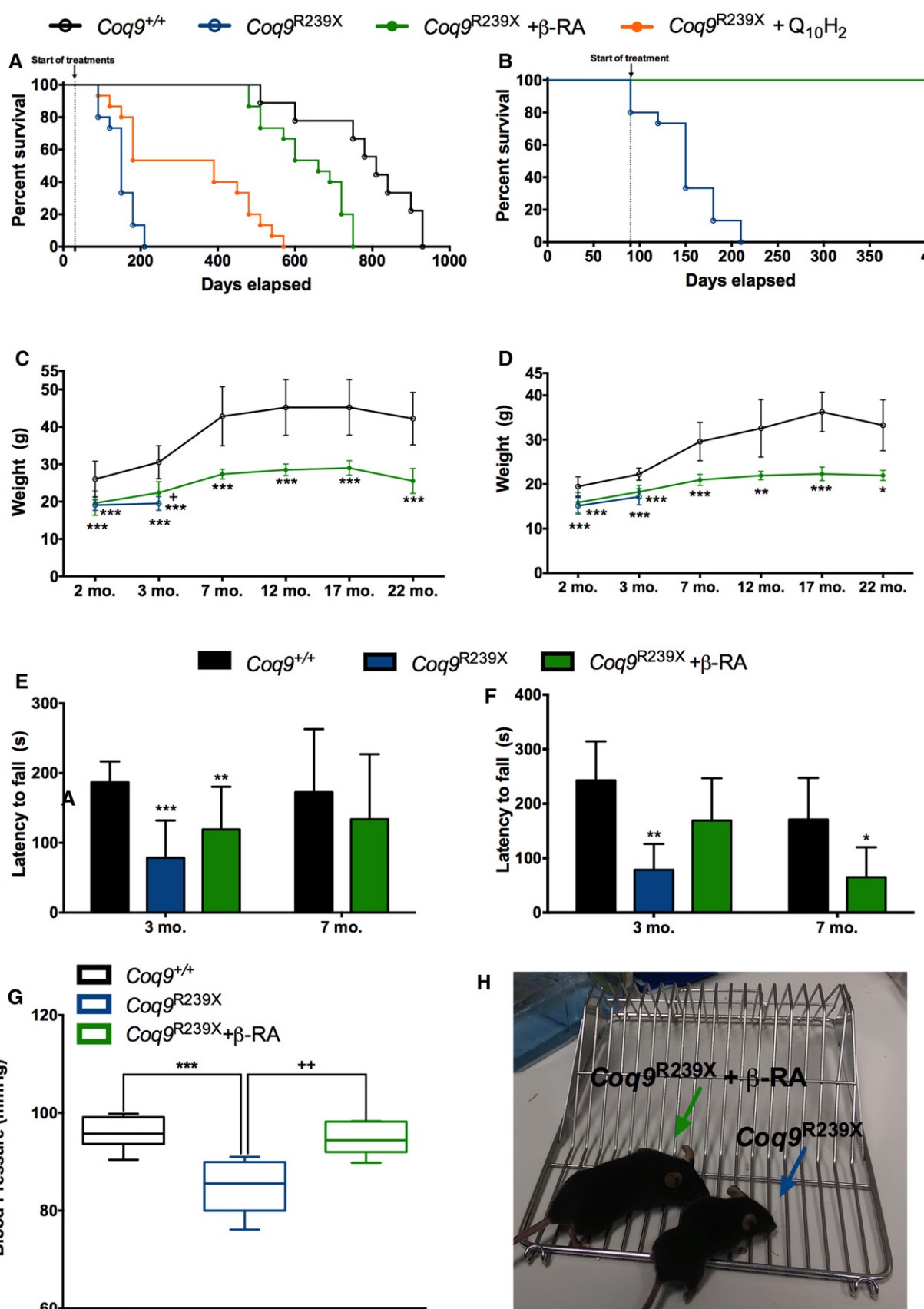

**Figure 1.**

Analysisusing the log-rank (Mantel-Cox) and the Gehan–Breslow–Wilcoxon tests shows significant differences ($P < 0.0001$ and $P = 0.0017$, respectively) between $Coq9^{+/+}$ mice, $Coq9^{R239X}$ mice, $Coq9^{R239X}$ mice treated with β-RA, and $Coq9^{R239X}$ mice treated with ubiquinol-10 (Fig 1A). Moreover, if the β-RA treatment started at 3 months of age (symptomatic period) instead of 1 month of age (asymptomatic period), 100% of the treated mice remained alive at 14 months of age (Fig 1B).

The striking increase in survival was in parallel of/accompanied by a phenotypic improvement. $Coq9^{R239X}$ mice treated with β-RA showed a slight increase in the body weight, although the values did not reach the wild-type levels (Fig 1C and D). The motor coordination tested by rotarod assay showed a decrease in the latency to fall in $Coq9^{R239X}$ mice compared to $Coq9^{+/+}$ mice. The treatment with β-RA significantly attenuated the motor phenotype in $Coq9^{R239X}$ mice, both

in males and females (Fig 1E and F). The treatment also normalized the blood pressure (Fig 1G), while the blood cells and hemoglobin levels did not change in the experimental groups (Appendix Fig S1). The general health improvement of $Coq9^{R239X}$ mice treated with β-RA was remarkable, clearly showing a healthier demeanor, compared to the untreated $Coq9^{R239X}$ mice (Fig 1H and Movie EV1).

**The treatment with β-RA induces a reduction in the histopathological signs of the encephalopathy**

Because of the striking effect of the treatment on the survival and health status of the $Coq9^{R239X}$ mice, we performed a histopathological evaluation of the brain, the main symptomatic tissue. The spongiform degeneration and reactive astrogliosis characteristic in the diencephalon (Fig 2B₁, E₁, H₁, and K₁) and

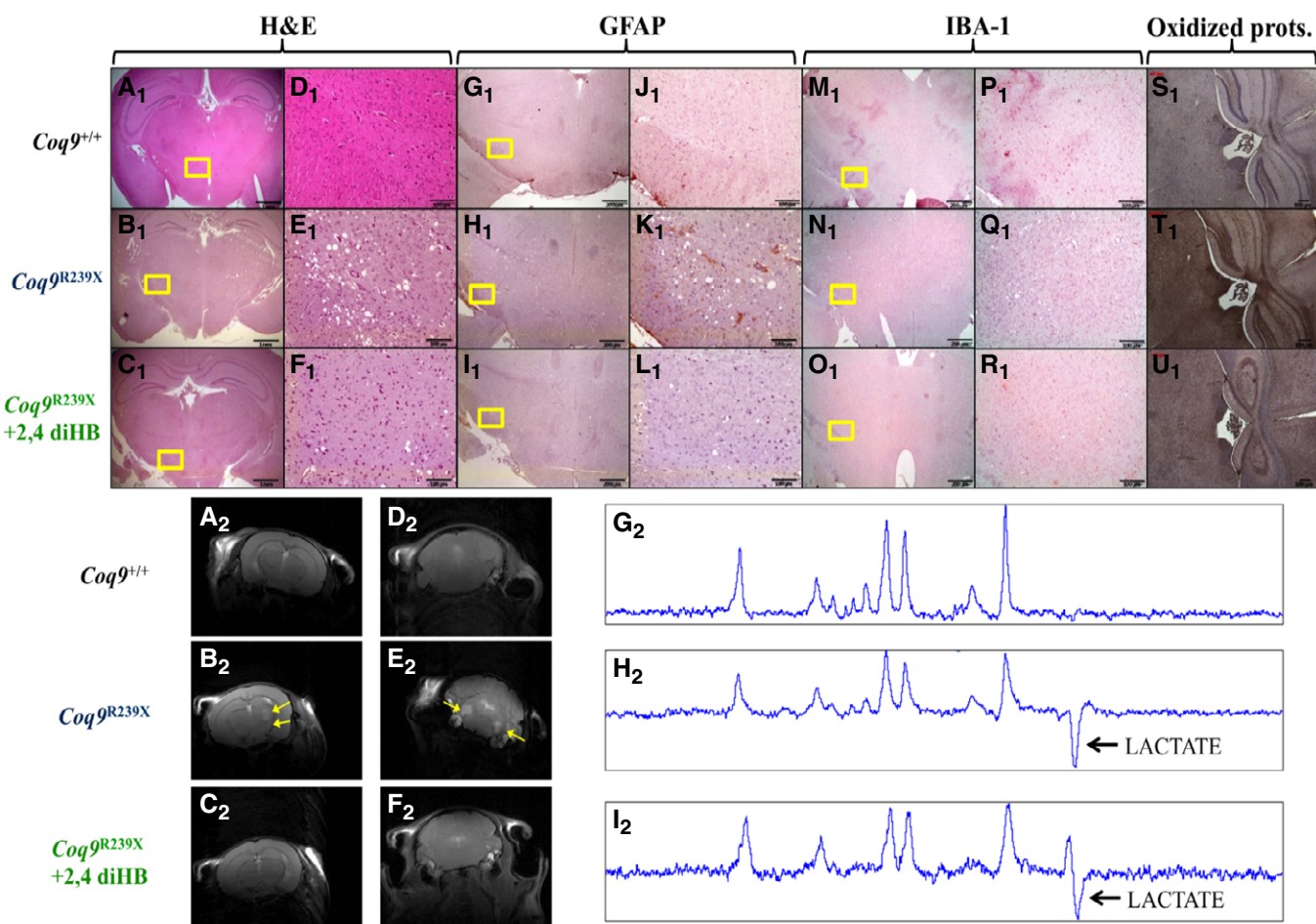

**Figure 2.  Pathological features in the brain of $Coq9^{R239X}$ mice after β-RA treatment.**

A–U    (A₁–F₁) H&E stain in the diencephalon of $Coq9^{+/+}$ mice (A₁ and D₁), $Coq9^{R239X}$ mice (B₁ and E₁), and $Coq9^{R239X}$ mice after β-RA treatment (C₁ and F₁). (G₁–L₁) Anti-GFAP stain in the diencephalon of $Coq9^{+/+}$ mice (G₁ and J₁), $Coq9^{R239X}$ mice (H₁ and K₁), and $Coq9^{R239X}$ mice after β-RA treatment (I₁ and L₁). (M₁–R₁) Anti-Iba1 stain in the diencephalon of $Coq9^{+/+}$ mice (M₁ and P₁), $Coq9^{R239X}$ mice (N₁ and Q₁), and $Coq9^{R239X}$ mice after β-RA treatment (O₁ and R₁). (S₁–U₁) Protein oxidation in the diencephalon of $Coq9^{+/+}$ mice (S₁), $Coq9^{R239X}$ mice (T₁), and $Coq9^{R239X}$ mice after β-RA treatment (U₁). (A₂–C₂) Magnetic Resonance Images of the diencephalon of $Coq9^{+/+}$ mice (A₂), $Coq9^{R239X}$ mice (B₂), and $Coq9^{R239X}$ mice after β-RA treatment (C₂). (D₂–F₂) Magnetic Resonance Images of the pons of $Coq9^{+/+}$ mice (D₂), $Coq9^{R239X}$ mice (E₂), and $Coq9^{R239X}$ mice after β-RA treatment (F₂). (G₂–I₂) Lactate peak in the brain of $Coq9^{+/+}$ mice (G₂), $Coq9^{R239X}$ mice (H₂), and $Coq9^{R239X}$ mice after β-RA treatment (I₂). Scale bars: 1 mm (A₁–C₁); 100 μm (D₁–F₁); 200 μm (G₁–I₁); 100 μm (J₁–L₁); 200 μm (M₁–O₁); 100 μm (P₁–R₁); 100 μm (S₁–U₁). The yellow arrows indicated the areas of increased T2 signal, which is characteristic of lesions in specific brain areas.

the pons (Appendix Fig S2) of the $Coq9^{R239X}$ mice (Garcia-Corzo *et al*, 2013), compared to $Coq9^{+/+}$ mice (Fig 2A$_1$, D$_1$, G$_1$, and J$_1$; Appendix Fig S2), almost disappeared after the β-RA treatment (Fig 2C$_1$, F$_1$, I$_1$, and L$_1$; Appendix Fig S2). The microglia distribution did not show any difference between the three experimental groups (Fig 2M$_1$–R$_1$; Appendix Fig S2). Moreover, the increased protein oxidation observed in the diencephalon of $Coq9^{R239X}$ mice (Fig 2T$_1$), compared to $Coq9^{+/+}$ mice (Fig 2S$_1$), was attenuated in $Coq9^{R239X}$ mice treated with β-RA (Fig 2U$_1$).

The improvements in the histopathological features were further corroborated by magnetic resonance imaging (MRI). Brain injuries were observed in the diencephalon and the pons of $Coq9^{R239X}$ mice (Fig 2B$_2$–E$_2$; Appendix Fig S2). In $Coq9^{+/+}$ mice and $Coq9^{R239X}$ mice treated with β-RA, however, we did not detect any sign of brain injury (Fig 2A$_2$, D$_2$, C$_2$ and F$_2$; Appendix Fig S2), suggesting that the treatment normalized the cerebral structure.

Nevertheless, the cerebral lactate levels, that are increased in $Coq9^{R239X}$ mice (Fig 2H$_2$) compared to $Coq9^{+/+}$ mice (Fig 2G$_2$), remained increased in $Coq9^{R239X}$ mice after the treatment (Fig 2I$_2$).

In the skeletal muscle, the cytochrome c oxidase (COX) and succinate dehydrogenase (SDH) stainings showed an increase in type II fibers in $Coq9^{R239X}$ mice. After the treatment with β-RA, the proportion of type I and type II fibers was normalized (Fig EV2).

### Transcriptomics profile reveals a reduction in the neuroinflammatory genes

Next, we carried out an RNA-Seq experiment on brainstems from the three experimental mice groups, e.g., $Coq9^{+/+}$, $Coq9^{R239X}$, and $Coq9^{R239X}$ treated with β-RA. The over $9 \times 10^9$ bases of the 101-bases sequencing reads aligned to 27,291 loci of the reference mouse genome. As Fig 3A shows, 298 genes were over-expressed in the $Coq9^{R239X}$ mice compared to the $Coq9^{+/+}$ mice, while 161 were under-expressed. On the other hand, 187 genes were over-expressed in the $Coq9^{R239X}$ mice compared to the $Coq9^{R239X}$ mice treated with β-RA, while 160 were under-expressed (Fig 3A). When comparing the pathways to which these differentially expressed genes belong, we observed a noticeably higher presence of genes belonging to inflammation signaling pathways in the $Coq9^{R239X}$ mice compared to the β-RA treated $Coq9^{R239X}$ or $Coq9^{+/+}$ mice (Fig 3B).

Of the 459 genes whose expression is significantly altered in $Coq9^{R239X}$ mice compared to $Coq9^{+/+}$, 27 were significantly altered in $Coq9^{R239X}$ mice treated with β-RA compared to $Coq9^{R239X}$ mice. Interestingly, among these 27 genes, the ones that were over-expressed in $Coq9^{R239X}$ mice, compared to the $Coq9^{+/+}$ mice, became under-expressed in $Coq9^{R239X}$ mice treated with β-RA, compared to $Coq9^{R239X}$ mice, and *vice versa* (Fig 3C). Thus, the levels of these 27 genes were normalized. Most of these genes showed higher expression in $Coq9^{R239X}$ mice compared to $Coq9^{+/+}$ mice, and they mainly have inflammation and immune-related functions. For example, the inflammatory genes *Bgn* (biglycan), *Ccl6* (chemokine C-C motif ligand 6), *Cst7* (cystatin F), *Ifi27l2a* (interferon, alpha-inducible protein 27 like 2A), Ifitm3 (interferon-induced transmembrane protein 3), *Itgax* (integrin alpha X), and *Vav1* are up-regulated in $Coq9^{R239X}$ mice and normalized after the treatment.

### β-RA does not directly act as anti-inflammatory agent

Because the transcriptomics profile uncovers a possible involvement of neuroinflammation in the therapeutic effect of β-RA, and since this molecule is an analogue of the anti-inflammatory salicylic acid, we tested whether β-RA may have a direct anti-inflammatory action. First, we incubated RAW cells with 1 μg/ml LPS, which induces a NF-κB-mediated inflammatory response, as it is shown by the induction of iNOS (Fig 3D) and the release of IL-1β, IL-2, and TNF-α (Fig 3E). The pre-incubation with β-RA, however, did not modify the cellular iNOS expression (Fig 3D) or the cytokine levels in the cells supernatants (Fig 3E). Second, we quantified the cytokine levels in the brainstem of the three animals' experimental groups. The levels of IL-1β, IL-2, IFN-γ, Gro-α, and TNF-α did not experience major changes (Fig 3F). Third, we tested whether β-RA may increase the survival in two models of neuroinflammation and early death due to mitochondrial Complex I deficiency, i.e., the zebrafish embryos treated with MPTP, which mimics Parkinson disease (Diaz-Casado *et al*, 2018); and the *Ndufs4* knockout mouse model, which mimics Leigh syndrome (Quintana *et al*, 2010). In both cases, the treatment with β-RA did not induce any change in the animal survival (Fig 3G and I) or the presence of malformations (Fig 3H). Because the inhibition of mTORC1 may mediate the anti-inflammatory actions of salicy-lates (Cameron *et al*, 2016), we also quantified the S6RP/S6R ratio as a marker of mTORC1 activity. Consistently with the other data, the β-RA did not modify the S6RP/S6R ratio in the brain (Appendix Fig S3A) and the kidneys (Appendix Fig S3B) of $Coq9^{R239X}$ mice treated with β-RA, suggesting that this molecule does not inhibit mTORC1 *in vivo*. Overall, these results do not support a direct role of β-RA in the repression of the typical neuroinflammatory pathways.

### β-RA improves mitochondrial bioenergetics in peripheral tissues of $Coq9^{R239X}$ mice

The persistent lactate peak observed in the brain spectroscopy after the β-RA treatment suggests that the treatment does not have any effect on mitochondrial bioenergetics in the brain. To confirm this premise, we evaluated the mitochondrial function in the brain, but also in the kidneys, muscle, and heart (tissues clinically important in CoQ deficiency syndrome) of $Coq9^{+/+}$ mice, $Coq9^{R239X}$ mice, and $Coq9^{R239X}$ mice treated with β-RA.

In the brain of the mutant mice, the activities of the CoQ-dependent mitochondrial Complexes I+III (CI+III) and CII+III, the levels of sulfide:quinone oxidoreductase (SQOR), the assembly of C-III into the supercomplexes (SC), and the mitochondrial oxygen consumption rate (OCR) were significantly decreased (Fig 4A–E) compared to the values of wild-type animals. The β-RA treatment did not induce any change in those parameters in the brain of $Coq9^{R239X}$ mice (Fig 4A–E). In the kidneys of the mutant mice, the activity of the CI+III, the levels of SQOR, and the OCR were significantly decreased (Fig 4F–J), compared to the values of wild-type animals. These changes were not due to a potential capability of β-RA to act as an electron carrier in the mitochondrial respiratory chain because the CI+III activity did not change after adding 50 μM β-RA into the reaction mix (Appendix Fig S4). The β-RA treatment partially

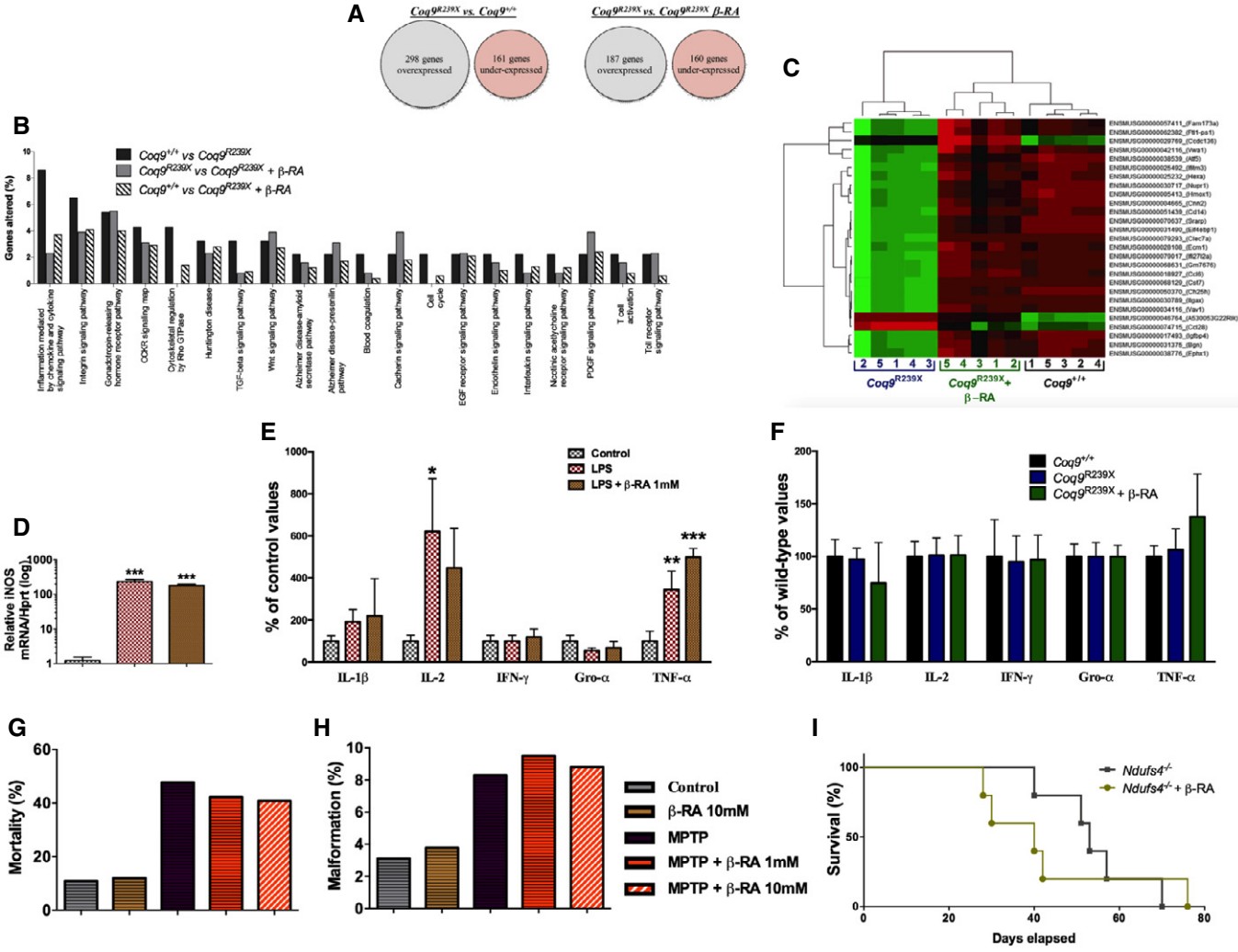

**Figure 3.  Effect of the β-RA treatment on genes expression profiles and neuroinflammatory-related features.**

A    Global differences in the levels of gene expression between experimental groups.

B    Only pathways with more than one gene significantly altered in *Coq9^R239X^* compared to *Coq9^+/+^* mice are considered for this figure. The y-axis represents the percentage of genes corresponding to a pathway in the sample of genes whose expression level is significantly altered in a comparison between mice samples.

C    Heatmap showing the comparative landscape of the expression level per sample of the 27 genes whose expression was altered by the mutation and normalized by the β-RA treatment. The column numbers correspond to the different samples and the row names are gene codes composed of the ENSEMBL gene ID and the gene symbol (between parentheses). The green means comparatively higher, red means comparatively lower, and black means no difference in expression level (n = 5 for each group).

D    iNOS expression levels in RAW cells after stimulation with LPS (n = 3 for each group).

E    Cytokine levels in the culture supernatant of RAW cells after stimulation with LPS (n = 3 for each group).

F    Cytokine levels in the brainstem of *Coq9^+/+^* mice, *Coq9^R239X^* mice, and *Coq9^R239X^* mice treated with β-RA (n = 7 for each group).

G, H  Mortality rate (G) and (H) percentage of malformations in zebrafish embryos treated with MPT and the effect of β-RA treatment.

I    Survival curve of *Ndufs4* knockout mice and *Ndufs4* knockout mice treated with β-RA [Log-rank (Mantel-Cox) test; n = 5 for each group; see Appendix Table S3 for exact *P*-values].

Data information: Data in panels (D–F) are expressed as mean ± SD. *P < 0.05; **P < 0.01; ***P < 0.001; LPS or LPS + β-RA versus control. +P < 0.05; LPS or LPS + β-RA versus control (one-way ANOVA with a Tukey's *post hoc* test; see Appendix Table S3 for exact *P*-values).

---

normalized the CI+III activity and induced a significant increase of SQOR levels and the OCR (Fig 4F–J). On the contrary, assembly of C-I into the supercomplexes did not change in any experimental group (Appendix Fig S5). In the skeletal muscle of the mutant mice, the activity of the CI+III was significantly decreased, while the CII+III activity did not change (Fig 4K and L), compared to the values of wild-type animals. The β-RA treatment normalized the

CI+III activity (Fig 4K). In the heart of *Coq9^R239X^* mice, the activities of the CI+III and CII+III, as well as the levels of SQOR, were significantly decreased (Fig 4M–O), compared to the values of wild-type animals. Again, the β-RA treatment partially normalized the CI+III, CII+III activities, and the levels of SQOR, although partially in this case (Fig 4M–O). Therefore, these results illustrate tissue-specific responses to β-RA treatment in the mitochondrial bioenergetics.

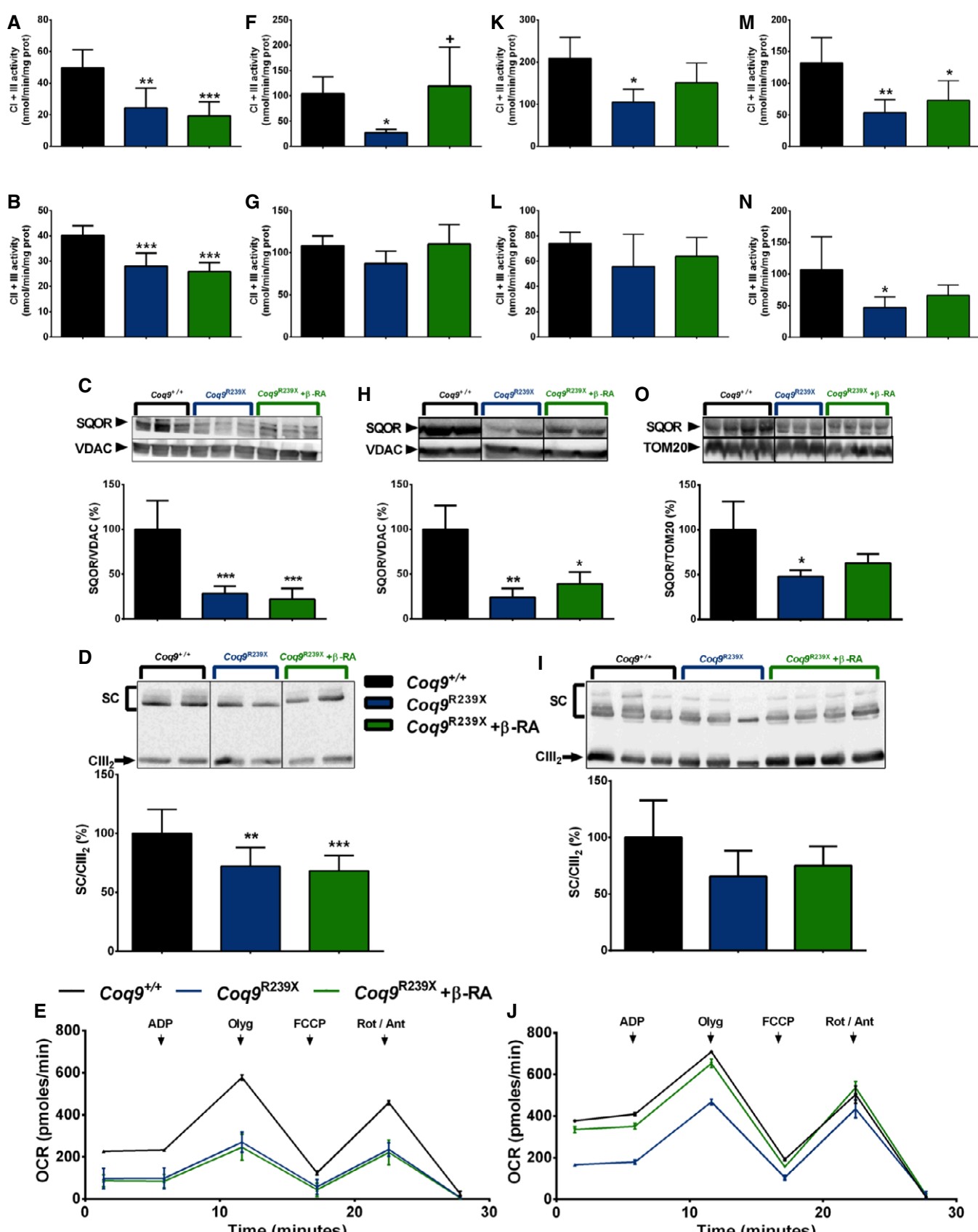

Figure 4.

**Figure 4.  Tissue-specific differences in mitochondrial function after β-RA treatment in *Coq9^R239X* mice.**

A–O    (A, F, K, M) CoQ-dependent Complex I+III activities in brain (A) and kidneys (F), skeletal muscle (K), and heart (M). (A): $Coq9^{+/+}$, n = 5; $Coq9^{R239X}$, n = 6; $Coq9^{R239X}$ after β-RA treatment, n = 7. (F): $Coq9^{+/+}$, n = 4; $Coq9^{R239X}$, n = 6; $Coq9^{R239X}$ after β-RA treatment, n = 4. (K): $Coq9^{+/+}$, n = 3; $Coq9^{R239X}$, n = 6; $Coq9^{R239X}$ after β-RA treatment, n = 3. (M): $Coq9^{+/+}$, n = 5; $Coq9^{R239X}$, n = 6; $Coq9^{R239X}$ after β-RA treatment, n = 6. (B, G, L, N) CoQ-dependent Complex II+III activities in brain (B) and kidneys (G), skeletal muscle (L) and heart (N). (B): $Coq9^{+/+}$, n = 5; $Coq9^{R239X}$, n = 6; $Coq9^{R239X}$ after β-RA treatment, n = 7. (G): $Coq9^{+/+}$, n = 5; $Coq9^{R239X}$, n = 6; $Coq9^{R239X}$ after β-RA treatment, n = 6. (L): $Coq9^{+/+}$, n = 5; $Coq9^{R239X}$, n = 6; $Coq9^{R239X}$ after β-RA treatment, n = 7. (N): $Coq9^{+/+}$, n = 6; $Coq9^{R239X}$, n = 6; $Coq9^{R239X}$ after β-RA treatment, n = 6. (C, H, O) Representative Western blot and quantitation of Western blot bands of SQOR in mitochondrial fractions of brain (C) and homogenates extracts from kidneys (H) and heart (O). (C): $Coq9^{+/+}$, n = 6; $Coq9^{R239X}$, n = 6; $Coq9^{R239X}$ after β-RA treatment, n = 6. (H): $Coq9^{+/+}$, n = 4; $Coq9^{R239X}$, n = 3; $Coq9^{R239X}$ after β-RA treatment, n = 3. (O): $Coq9^{+/+}$, n = 4; $Coq9^{R239X}$, n = 3; $Coq9^{R239X}$ after β-RA treatment, n = 4. (D, I) Blue-native gel electrophoresis (BNGE) followed by C-III immunoblotting analysis of mitochondrial supercomplexes in brain (D) and kidneys (I). (D): $Coq9^{+/+}$, n = 9; $Coq9^{R239X}$, n = 10; $Coq9^{R239X}$ after β-RA treatment, n = 11. (I): $Coq9^{+/+}$, n = 5; $Coq9^{R239X}$, n = 6; $Coq9^{R239X}$ after β-RA treatment, n = 7. (E, J) Mitochondrial oxygen consumption rate (represented as State 3o, in the presence of ADP and substrates) in brain (E) and kidneys (J). The data represent three technical replicates and the figures are representative of three biological replicates. $Coq9^{+/+}$ mice, $Coq9^{R239X}$ mice, and $Coq9^{R239X}$ mice after β-RA treatment were included in all graphs. Data are expressed as mean ± SD. *$P < 0.05$; **$P < 0.01$; ***$P < 0.001$; $Coq9^{R239X}$ or $Coq9^{R239X}$ after β-RA treatment versus $Coq9^{+/+}$. +$P < 0.05$; $Coq9^{R239X}$ versus and $Coq9^{R239X}$ after β-RA treatment (one-way ANOVA with a Tukey's *post hoc* test; see Appendix Table S4 for exact *P*-values).

Source data are available online for this figure.

## β-RA reduces DMQ₉ in peripheral tissues of *Coq9^R239X* mice

$Coq9^{R239X}$ mice present widespread deficiency of $CoQ_9$ (major CoQ form in rodents) and $CoQ_{10}$ (minor CoQ form in rodents). Because COQ9 protein is needed for the COQ7 hydroxylation reaction, $Coq9^{R239X}$ mice also show accumulation of demethoxyubiquinone-9 ($DMQ_9$), the substrate of COQ7 (Garcia-Corzo et al, 2013). To test whether the general health improvement after β-RA treatment was due to the effects of this compound on CoQ biosynthesis, and to check whether the tissue-specific differences in mitochondrial bioenergetics were due to variances in the CoQ metabolism, we measured the levels of $CoQ_9$, $CoQ_{10}$ and $DMQ_9$ in the most relevant tissues. As previously reported, the levels of $CoQ_9$ (Fig 5A, E, I, M and Q) and $CoQ_{10}$ (Fig 5B, F, J, N and R) were severely decreased in the brain (Fig 5A and B), kidneys (Fig 5E and F), liver (Fig 5I and J), skeletal muscle (Fig 5M and N), and heart (Fig 5Q and R) of $Coq9^{R239X}$ mice. At the same time, the levels of $DMQ_9$ were significantly increased in the same tissues (Fig 5C, G, K, O and S) and, therefore, the $DMQ_9/CoQ_9$ ratios were significantly increased (Fig 5D, H, L, P, and T) in $Coq9^{R239X}$ mice. After the treatment with β-RA, the levels of $CoQ_9$ slightly increased only in the kidneys (Fig 5E), but not in the other tissues (Fig 5A, I, M, and Q) of the mutant mice. $CoQ_{10}$ also showed a trend toward increased levels in the kidneys of $Coq9^{R239X}$ mice after β-RA treatment (Fig 5F), although in the other tissues the levels of $CoQ_{10}$ did not change after β-RA treatment (Fig 5B, J, N and R). Importantly, the levels of $DMQ_9$ and, consequently, the $DMQ_9/CoQ_9$ ratios were significantly decreased in the kidneys (Fig 5G and H), liver (Fig 5K and L), skeletal muscle (Fig 5O and P), and heart (Fig 5S and T) of $Coq9^{R239X}$ mice treated with β-RA, suggesting that β-RA exerts an influence in the CoQ biosynthetic pathway. In the brain, however, the β-RA treatment did not induce changes in the levels of $DMQ_9$ or $DMQ_9/CoQ_9$ ratio (Fig 5C and D). Taken together, these results suggest that the improvement of mitochondrial bioenergetics occurs only in the tissues with decreased $DMQ_9/CoQ_9$ ratio after the treatment.

To check whether the levels of $CoQ_9$ and $DMQ_9$ can be further modified during the time of the treatment or with a higher dose of β-RA, we also quantified the levels of both quinones in tissues of mice after 9 months of treatment or with the double dose of β-RA. The results showed that the levels of $DMQ_9$ levels and the $DMQ_9/CoQ_9$ ratio were decreased in these two other experimental variants (Table EV1 and Appendix Figs S6 and S7). However, no

major differences were found in the results obtained under three experimental situations.

Because the tissue-specific differences observed in $DMQ_9/CoQ_9$ ratio after β-RA treatment could be due to the bioavailability of this compound in the tissues after oral supplementation, we quantified the levels of β-RA in $Coq9^{R239X}$ mice after oral supplementation and compared the data to those obtained after acute intraperitoneal (i.p.) administration. The compound was detected in all the examined tissues, including the brain, by both oral and i.p. administration (Table EV2 and Appendix Fig S8). The highest values of β-RA were found in plasma and kidneys, while the results in brain and liver were similar (Table EV2 and Appendix Fig S8). In the non-treated animals, β-RA was not detected, except in the kidney of one animal (Table EV2 and Appendix Fig S8). In order to see whether the effect observed after the treatment was due to β-RA and not to any other metabolite derived from it, we developed an UPLC-MS screening experiment. The other hydroxybenzoic acid detected in the samples was the 4-HB, but no statistic differences were found between the experimental groups (Appendix Table S1). Therefore, the bioavailability of β-RA does not completely explain the tissue-specific differences in CoQ metabolism since brain and liver showed similar levels of β-RA.

## The effect of β-RA over CoQ biosynthetic pathway is mediated by a mild stabilization of the Complex Q

To try to understand the mechanism involved in the reduction in the levels of $DMQ_9$ observed after β-RA treatment, we quantified the levels of main CoQ biosynthetic proteins. The levels of PDSS2 and COQ2, which are not part of Complex Q, did not experience major changes in the brain (Appendix Fig S9A and B), kidneys (Appendix Fig S9C and D), and heart (Appendix Fig S9F and G) of the three experimental groups, and only a decrease in the COQ2 levels was appreciated in the kidneys of $Coq9^{R239X}$ mice (Appendix Fig S9D). The levels of COQ4, COQ5, COQ6, COQ7, and COQ8A, which are components of the Complex Q, were consistently decreased in the brain (Fig 6A–E), kidneys (Fig 6F–J), and heart (Fig 6K–O) of $Coq9^{R239X}$ mice compared to the same tissues in $Coq9^{+/+}$ mice. Interestingly, the treatment with β-RA induced a trend toward increase on the levels of COQ4, COQ5, COQ6, and COQ8A in the kidneys (Fig 6F, G, H and J) and heart (Fig 6K, L, M and O), but not in the brain (Fig 6A, B, C and E). On the contrary,

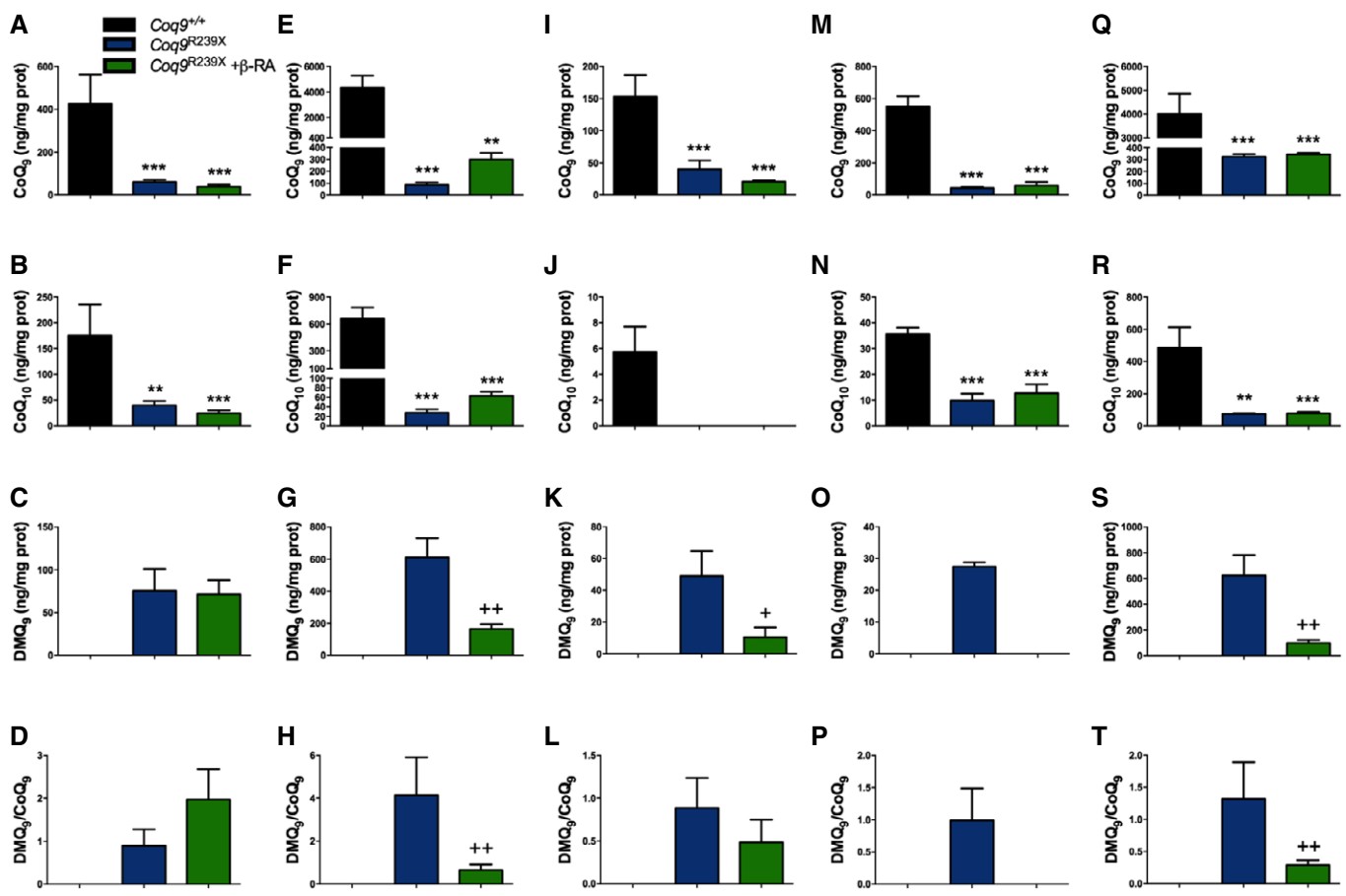

**Figure 5.  Tissue-specific differences in the levels of CoQ₉, CoQ₁₀, and DMQ₉, and in the DMQ₉/CoQ₉ ratio after β-RA treatment in *Coq9^R239X* mice.**

A–T    Levels of CoQ₉ (A, E, I, M, Q), CoQ₁₀ (B, F, J, N, R), and DMQ₉ (C, G, K, O, S), as well as and DMQ₉/CoQ₉ ratio (D, H, L, P, T) in brain (A–D), kidney (E–H), liver (I–L), skeletal muscle (M–P), and heart (Q–T) of *Coq9^+/+* mice, *Coq9^R239X* mice, and *Coq9^R239X* mice after β-RA treatment. Data are expressed as mean ± SD. **$P < 0.01$; ***$P < 0.001$; *Coq9^R239X* or *Coq9^R239X* after β-RA treatment versus *Coq9^+/+*. ⁺$P < 0.05$; ⁺⁺$P < 0.01$; *Coq9^R239X* versus *Coq9^R239X* after β-RA treatment (one-way ANOVA with a Tukey's *post hoc* test or *t*-test; $n = 6$ for each group; see Appendix Table S5 for exact *P*-values). Note that DMQ9 was undetectable in tissues from *Coq9^+/+* mice as well as in muscle of *Coq9^R239X* mice treated with β-RA.

the levels of COQ7, which physically interacts with COQ9, were not modified by the β-RA treatment in any of the three tissues (Fig 6D, J and T). Overall, these results suggest that β-RA may induce the stabilization of the Complex Q, resulting in a decrease in the DMQ₉/CoQ₉ ratio.

**The levels of DMQ₉ contribute to the disease phenotype**

The results of the previous sections suggest that the DMQ₉/CoQ₉ ratio is a key factor in the disease phenotype. To validate those results, we calculated the DMQ₉/CoQ₉ ratio in the tissues of two additional experimental models: (i) the *Coq9^Q95X* mouse model, which have widespread CoQ deficiency without compromising the lifespan (Luna-Sanchez *et al*, 2015); and (ii) the *Coq9^R239X* mice treated with ubiquinol-10, which show an increase in the levels of CoQ₁₀ (Garcia-Corzo *et al*, 2014), resulting in an increase in the lifespan (Fig 1A); yet, this increased lifespan is not comparable to the one observed under β-RA treatment (Fig 1A). Our results show that, at 1 month of age, the DMQ₉/CoQ₉ ratios in the tissues

of *Coq9^Q95X* mice were significantly lower than in the tissues of *Coq9^R239X* mice (Fig 7A and B; Appendix Fig S10). At 3 months of age, the DMQ₉ almost disappeared in the tissues of *Coq9^Q95X*, but not in *Coq9^R239X* mice (Fig 7C and D; Appendix Fig S10). In *Coq9^R239X* mice, the treatment with ubiquinol-10 did not affect the levels of DMQ₉ and, therefore, the DMQ₉/CoQ₉ ratios were still high in all tissues, while the CoQ₁₀ was increased (Fig 7E and F; Appendix Fig S11). Therefore, those results corroborate the correlation between DMQ₉/CoQ₉ ratio and the severity of the disease phenotype.

**Steroid hormones or FGF21 do not mediate the tissues communication under CoQ deficiency**

Because our data suggest that the modification of CoQ metabolism in peripheral tissues induces a reduction in the brain pathology in *Coq9^R239X* mice treated with β-RA, we tried to identify a circulating signal molecule that may provide this tissue communication (e.g., kidney–brain, muscle–brain, heart–brain, or

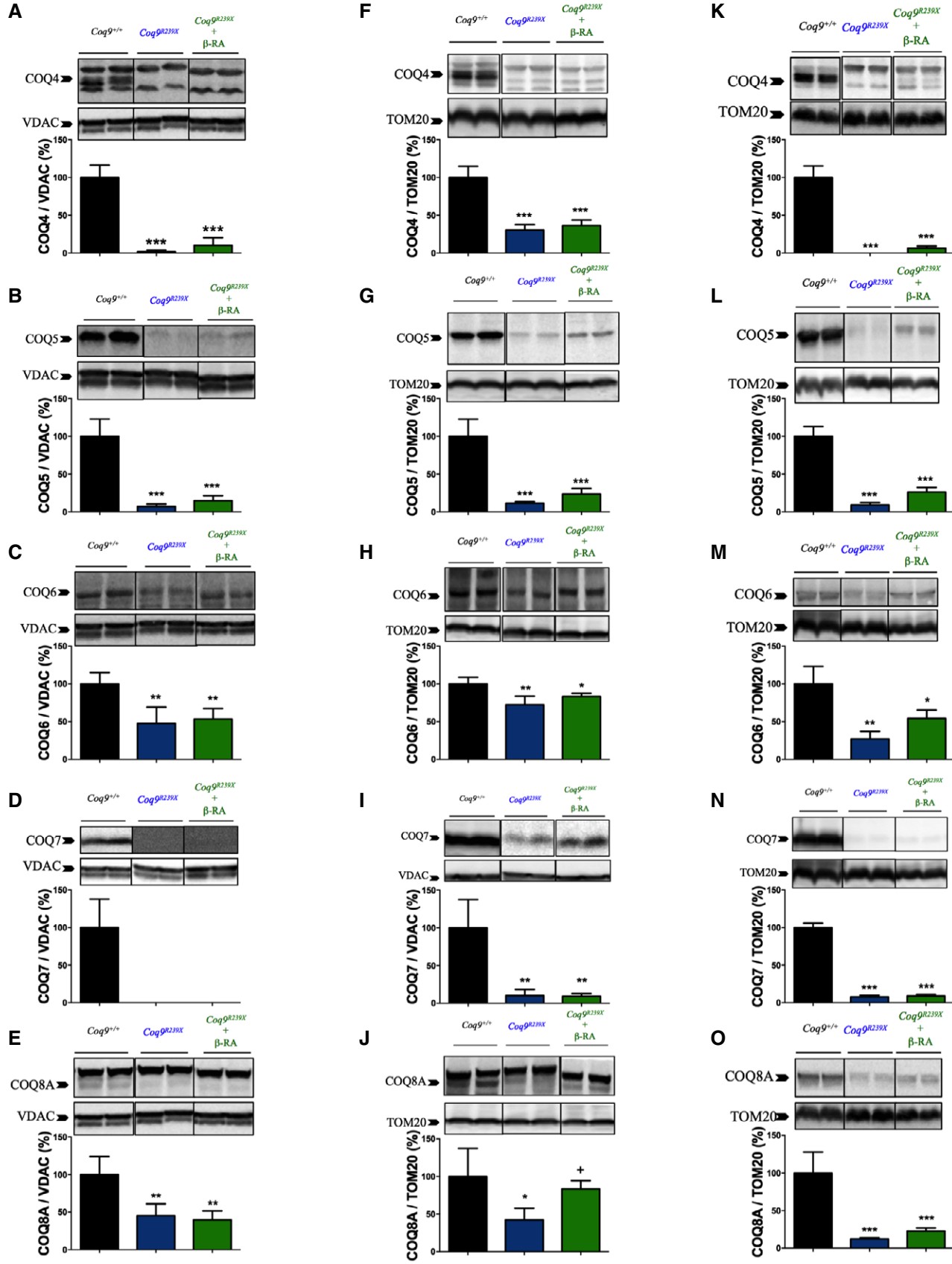

Figure 6.

◀

**Figure 6.  Effect of β-RA treatment in the tissue levels of CoQ biosynthetic proteins.**

A–O  Representative images of Western blots of the CoQ biosynthetic proteins COQ4, COQ5, COQ6, COQ7, and COQ8A and the quantitation of the protein bands in the brain (A–E), kidneys (F–J), and heart (K–O) of $Coq9^{+/+}$, $Coq9^{R239X}$, and $Coq9^{R239X}$ mice treated with β-RA. $Coq9^{+/+}$ mice, $Coq9^{R239X}$ mice, and $Coq9^{R239X}$ mice after β-RA treatment were included in all graphs. Data are expressed as mean ± SD. *$P < 0.05$; **$P < 0.01$; ***$P < 0.001$; $Coq9^{R239X}$ or $Coq9^{R239X}$ after β-RA treatment versus $Coq9^{+/+}$. +$P < 0.05$; $Coq9^{R239X}$ versus and $Coq9^{R239X}$ after β-RA treatment (one-way ANOVA with a Tukey's *post hoc* test; n = 5 for each group; see Appendix Table S6 for exact *P*-values).

Source data are available online for this figure.

liver–brain). With that purpose, we quantified the steroid hormones, since some steps of their biosynthesis are localized in mitochondria; and FGF21, a recently identified biomarker for mitochondrial myopathies (Suomalainen *et al*, 2011). The results did not show any differences in the levels of corticosterone between the three experimental groups, both in males and females (Appendix Fig S12A and B). In male $Coq9^{R239X}$ mice, the levels of androstenedione and testosterone were significantly lower than those in male $Coq9^{+/+}$ mice (Appendix Fig S9C and D). After β-RA treatment, the levels of both hormones were still lower than those in control animals (Appendix Fig S12C and D).

The levels of FGF21 in the target organ, the brain, were similar in $Coq9^{+/+}$ and $Coq9^{R239X}$ mice, while a slight increase was detected in the precursor form in $Coq9^{R239X}$ mice treated with β-RA,

compared to $Coq9^{R239X}$ mice (Appendix Fig S12E). In the liver, the main secretory organ of FGF21, the levels of FGF21 (both the precursor and the mature form) were similar in the three experimental groups (Appendix Fig S12F).

# Discussion

The treatment of primary or secondary mitochondrial diseases is, in most cases, still limited to a palliative care. In this study, we identify a novel treatment for primary CoQ deficiency. The treatment is based on the administration of β-RA, which reduces the molecular, histopathologic, and clinical signs of mitochondrial encephalopathy in a mouse model of encephalopathy associated with CoQ

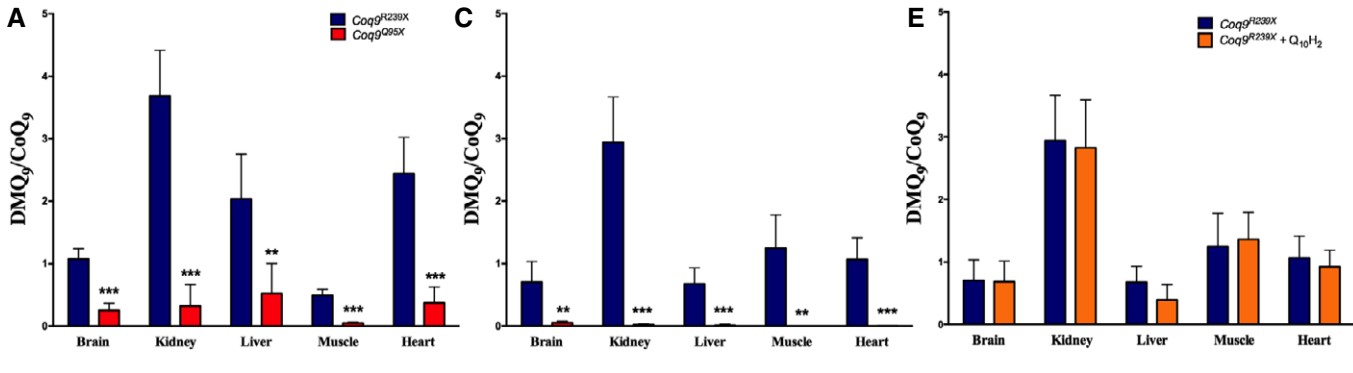

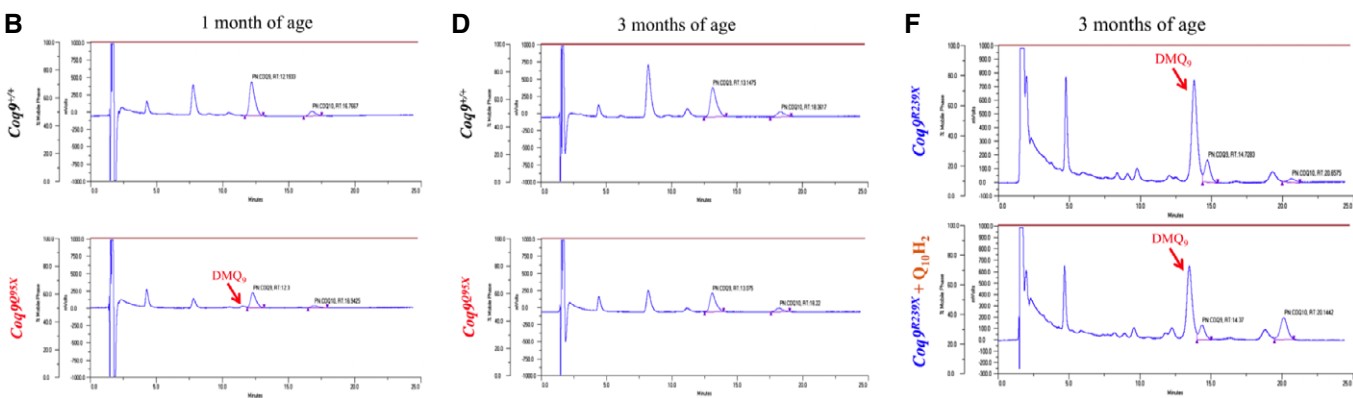

**Figure 7.  DMQ$_9$/CoQ$_9$ ratio in other experimental conditions.**

A–F  (A) DMQ$_9$/CoQ$_9$ ratio in tissues from $Coq9^{R239X}$ and $Coq9^{Q95X}$ mice at 1 month of age. (C) DMQ$_9$/CoQ$_9$ ratio in tissues from $Coq9^{R239X}$ and $Coq9^{Q95X}$ mice at 3 months of age. (E) DMQ$_9$/CoQ$_9$ ratio in tissues from $Coq9^{R239X}$ mice and $Coq9^{R239X}$ mice treated with ubiquinol-10 during 2 months. (B, D, F) Representative chromatographs showing the different quinones in the kidneys. Data in panels (A, C, E) are expressed as mean ± SD. **$P < 0.01$; ***$P < 0.001$; $Coq9^{Q95X}$ versus $Coq9^{R239X}$ (*t*-test; n = 5 for each group; see Appendix Table S7 for exact *P*-values).

deficiency. The remarkable increase of survival after β-RA treatment was superior to that observed after ubiquinol-10 treatment, and the therapeutic effect was succeeded even if the treatment started in a symptomatic period. The striking increase in animal survival and the general health improvement after the β-RA treatment was achieved even though the levels of CoQ and mitochondrial function were not increased in the brain, which is the most relevant tissue for the development of the disease. In peripheral tissues, however, β-RA induced a decrease in the $DMQ_9/CoQ_9$ ratio, resulting in a bioenergetics improvement that may exert a key influence on the brain pathology. Accordingly, we provide additional data about the $DMQ_9/CoQ_9$ ratio in the $Coq9^{Q95X}$ mouse model and in $Coq9^{R239X}$ mice treated with ubiquinol-10, and both cases support the importance of the $DMQ_9/CoQ_9$ ratio in the progression of the clinical symptoms associated with CoQ deficiency syndrome.

The initial hypothesis for this study was that β-RA could be used in the CoQ biosynthetic pathway and bypasses the defect in COQ9-COQ7, increasing the endogenous levels of CoQ (Luna-Sanchez et al, 2015; Pierrel, 2017). However, our results show that the bypass effect would be limited because only the kidneys, the tissue with higher levels of β-RA, showed a moderate increase in the levels of CoQ in the mutant mice after the treatment. Nevertheless, the levels of $DMQ_9$ were significantly decreased not only in the kidneys but also in the skeletal muscle, liver, and heart. Those results hint that the CoQ biosynthetic pathway might use β-RA as a substrate whose Km is probably higher than the natural substrate, 4-HB, and for that reason the reduction of the $DMQ_9$ levels is not accompanied by an increase of $CoQ_9$ in tissues where the β-RA does not reach a threshold (Pierrel, 2017). Also, the stabilization of the Complex Q may contribute to this effect, since the levels of COQ4, COQ5, COQ6, and COQ8A were slightly increased in the kidneys and heart after β-RA treatment. Consistent with those results, 4-HB or vanillic acid induced an increase of the CoQ biosynthetic protein in vitro (Herebian et al, 2017b,c). Moreover, we must also consider that the β-RA administration was able to increase the levels of $CoQ_9$ in renal and cardiac mitochondria, as well as in muscle tissue in a conditional Coq7 KO model, which lacks COQ7 protein (Wang et al, 2015, 2017). Nevertheless, the cerebral $CoQ_9$ levels after β-RA supplementation were not determined in the conditional Coq7 KO mice (Wang et al, 2015, 2017). Because $Coq9^{R239X}$ mice has reduced levels not only of COQ7 but also of COQ4, COQ5, COQ6, and COQ8A (Luna-Sanchez et al, 2015), it is also possible that the limitation of the β-RA to generally increase the $CoQ_9$ levels in $Coq9^{R239X}$ mice could be due to the disruption of the Complex Q observed in this mouse model. The combination of both arguments could explain the lack of effect over the CoQ biosynthesis pathway in the brain. In either case, our data on the $Coq9^{R239X}$ mouse model together with those obtained in the Coq7 conditional KO model indicate that it is possible to influence the CoQ biosynthesis in vivo with the use of the appropriated 4-HB analog(s). These results partially corroborate previous data in vitro using mutant yeasts and human skin fibroblasts derived from patients with primary CoQ deficiency (Ozeir et al, 2011; Xie et al, 2012; Doimo et al, 2014; Freyer et al, 2015; Luna-Sanchez et al, 2015; Herebian et al, 2017b,c; Pierrel, 2017; Wang et al, 2017).

While the levels of $CoQ_9$ and $CoQ_{10}$ were doubled in the kidneys of $Coq9^{R239X}$ mice after the treatment, the levels reached after the treatment were barely 5% of the wild-type levels. The reduction in

the $DMQ_9$ levels after the therapy, and as a consequence, the $DMQ_9/CoQ_9$ ratio, was more important in terms of absolute values, as well as more consistent because the response was similar in all peripheral tissues. Importantly, the decrease in $DMQ_9/CoQ_9$ ratio in peripheral tissues (like kidneys, heart, and muscle) was enough to increase the CoQ-dependent complexes activities up to normalizing the mitochondrial respiration. Therefore, the effect of the β-RA treatment in the function of the renal, muscular, and cardiac mitochondria must be due to the reduction of the $DMQ_9/CoQ_9$ ratio. While DMQ is not commercial available to develop more direct experiments, Yang et al (2011) were able to exchange the quinone pools between the mitochondria from wild-type and Clk-1 (= Coq7) mutated worms, demonstrating that $DMQ_9$ was the responsible of the inhibition of the electron transfer from complex I to ubiquinone in the mitochondrial respiratory chain, probably due to the competition of CoQ and DMQ for the same binding site, together with the inability of DMQ to transfer electrons (Arroyo et al, 2006). Our results in mice suggest that DMQ could also inhibit the electron transfer from complex II to ubiquinone, a fact that was not observed in worms (Yang et al, 2011). Those differences may reflect the diverse structure and functionality of the mitochondria in different organisms, tissues, and cell types (Vafai & Mootha, 2012). Thus, our study points out the importance of the $DMQ_9/CoQ_9$ ratio in the disease phenotype, a fact that is further supported by several evidences: (i) The $Coq9^{Q95X}$ mouse model has low levels of $DMQ_9$ and higher levels of CoQ biosynthetic proteins (compared to the $Coq9^{R239X}$ muse model), and the lifespan is not compromised; (ii) the treatment with ubiquinol-10 in $Coq9^{R239X}$ mice does not change the $DMQ_9$ levels and its therapeutic effects are significantly lower than those after β-RA treatment; (iii) the reduction in the $DMQ_9$ levels in the Coq7 conditional KO mice after β-RA therapy may also contribute to the increased survival observed in this mouse model, although the phenotype of this mouse model is not clear (Wang et al, 2015); and (iv) patients with high levels of $DMQ_{10}$ in the samples used for the diagnostic (muscle and/or skin fibroblasts) are associated to severe clinical presentations (Duncan et al, 2009; Freyer et al, 2015; Danhauser et al, 2016; Wang et al, 2017; Smith et al, 2018).

Because the levels of $CoQ_9$ and $DMQ_9$ did not change in the brain of the $Coq9^{R239X}$ mice after the treatment, this tissue did not experience any improvement in the mitochondrial function after β-RA therapy. However, even if the treatment does not have any detectable effect at the cerebral mitochondria, the $Coq9^{R239X}$ mice showed a clear and profound reduction in the spongiform degeneration, reactive astrogliosis, and oxidative damage, leading to an absence of brain injuries after the β-RA therapy. Those effects are, therefore, independent of the $CoQ_9$ and $DMQ_9$ levels in the brain. Thus, two complementary hypotheses could explain the therapeutic effects of β-RA in the brain of $Coq9^{R239X}$ mice. First, β-RA may have CoQ-independent functions with therapeutic potential for mitochondrial encephalopathies. The transcriptomics data showing a decrease in the expression levels of some inflammatory genes, and the fact that β-RA is a structural analogue of salicylic acid (Gomez-Guzman et al, 2014), would support this hypothesis. However, our experimental data do not support this hypothesis because: (i) The microglia distribution and the cytokines levels did not experience major changes between $Coq9^{R239X}$ mice and $Coq9^{R239X}$ mice treated with β-RA; (ii) β-RA did not reduce iNOS expression or cytokines release in RAW

cells stimulated with LPS; (iii) β-RA did not have therapeutic effects in pharmacological or genetic models of neuroinflammation, e.g., the MPTP-induced zebrafish model of Parkinson disease and the *Ndufs4* mouse model of Leigh syndrome; and (iv) β-RA did not inhibit mTORC1 *in vivo*. Thus, the changes in the expression levels of some inflammatory genes may reflect the reduction in the astrogliosis. In fact, some genes associated with chemokine (e.g., C-C Motif Chemokine Ligands family) and cytokine signaling (e.g., *Cd14*, *Ifitm3*, *Hmox1*), immune response (*Nurp1*), cell cycle (e.g., *Eif4bpp1*), cell adhesion (e.g., *Ecm1*, *Igfb4*), and lipid metabolism (e.g., *Ch25*) have been also associated to reactive astrogliosis (Zamanian *et al*, 2012; Colombo & Farina, 2016) and, therefore, the reduction observed in these genes could be due to the reduction in the astrogliosis. Our data also suggest that, similarly to other mitochondrial encephalopathies (Lax *et al*, 2012; Tegelberg *et al*, 2017), the neuroinflammation associated with microglia activation is not an important factor in the mitochondrial encephalopathy associated with CoQ deficiency.

The second hypothesis is that the reduction in $DMQ_9/CoQ_9$ ratio in peripheral tissues and the subsequent improvement of mitochondrial bioenergetics may provide some tissue–brain cross-talk in order to reduce the astrogliosis, spongiosis, and its associated brain injury. While we were not able to reveal how the peripheral tissues communicate with the brain in this situation, a recent study about gene therapy in a mouse model of Leigh syndrome also suggests that the correction of the gene defect in peripheral tissues is important to reduce brain pathology (Di Meo *et al*, 2017). The studies about the kidney–brain, heart–brain, or liver–brain cross-talk, as well as the presence of a gut–brain axis, would support the concept of an influence of peripheral tissues in the brain pathology, and reactive astrogliosis in particular (Mayo *et al*, 2014; Butterworth, 2015; Rothhammer *et al*, 2016, 2017; Miranda *et al*, 2017; Thackeray *et al*, 2018).

In conclusion, our study shows that β-RA is a powerful therapeutic agent for the mitochondrial encephalopathy due to CoQ deficiency, a disease that is increasingly diagnosed with exome sequencing, with an estimation of more the 120,000 cases worldwide (Hughes *et al*, 2017). The therapeutic results are greater than those obtained by the classical oral CoQ supplementation because of the decrease of the DMQ/CoQ ratio. Therefore, β-RA should be preferentially considered for the treatment of human $CoQ_{10}$ deficiency with accumulation of $DMQ_{10}$, as it has been reported in patients with mutations in *COQ9*, *COQ7*, or *COQ4* (Duncan *et al*, 2009; Freyer *et al*, 2015; Danhauser *et al*, 2016; Herebian *et al*, 2017b; Wang *et al*, 2017; Smith *et al*, 2018) but also in cells under siRNA knockdown of *COQ3*, *COQ5* and *COQ6* (Herebian *et al*, 2017b). Similar principles could be applied for 3,4-hydroxybenzoic acid, vanillic acid, 2-methyl-4-hydroxybenzoic acid, or 2,3-dimethoxy-4-hydroxybenzoic acid in the cases of mutations in *COQ6, COQ5,* or *COQ3* (Heeringa *et al*, 2011; Yoo *et al*, 2012; Ribas *et al*, 2014; Gigante *et al*, 2017; Herebian *et al*, 2017a; Pierrel, 2017), respectively. Furthermore, 4-HB analogs may provide therapeutic effects in other conditions, e.g., mitochondrial disorders, since secondary CoQ deficiency due to decreased levels of CoQ biosynthetic proteins have been recently reported in various mouse models of mitochondrial diseases (Kuhl *et al*, 2017); or metabolic diseases due to insulin resistance since secondary CoQ deficiency due to decreased levels of CoQ biosynthetic proteins have been recently described in

*in vitro* insulin resistance models and adipose tissue from insulin-resistant humans (Fazakerley *et al*, 2018). To apply the β-RA treatment into the clinic, the safety and dose–response studies included in a clinical trial should be first developed. Nevertheless, β-RA is a natural compound that is used as a flavor by the food industry (Adams *et al*, 2005), and its administration in wild-type mice at high doses does not have detrimental effects on the survival (Wang *et al*, 2015). Therefore, the safety study has a good prognosis and would allow the use of this compound at least as a compassionate option in cases of severe CoQ deficiencies due to mutations in *COQ9* or *COQ7*.

# Materials and Methods

### Experimental design

$Coq9^{+/+}$, $Coq9^{R239X}$, $Ndufs4^{+/+}$, and $Ndufs4^{-/-}$ mice were used in the study. All mice have a mix of C57BL/6N and C57BL/6J genetic background. Mice were housed in the Animal Facility of the University of Granada under an SPF zone with lights on at 7:00 AM and off at 7:00 PM. Mice had unlimited access to water and rodent chow. The $Coq9^{R239X}$ mouse model (http://www.informatics.jax.org/allele/key/829271) was previously generated and characterized (Garcia-Corzo *et al*, 2013). β-RA was incorporated in the chow at a concentration of 1%, which provides a dose similar to the one (1 g/kg b.w./day) that was therapeutically successful in the *Coq7* conditional KO model (Wang *et al*, 2015). For a particular experiment, we doubled the dose by adding β-RA into the drinking water at a concentration of 0.5%, neutralizing the water with sodium bicarbonate. Also, a pilot survival study was done with β-RA in *Ndufs4* knockout mice, which show microgliosis and neuroinflammation (Quintana *et al*, 2010).

The motor coordination was evaluated by rotarod test at different months of age. Systolic blood pressure (SBP) and heart rate (HR) were measured in conscious, prewarmed, and restrained mice by tail-cuff plethysmography (Digital Pressure Meter LE 5001, Letica S.A., Barcelona, Spain) as described previously (Gomez-Guzman *et al*, 2014; Luna-Sanchez *et al*, 2017).

The therapeutic potential of β-RA was also assessed in zebrafish embryos treated with the neurotoxin 1-methyl-4-phenyl-1,2,3,6-tetrahydropyridine (MPTP), which is used to induce key features of Parkinson disease, including CI inhibition and neuroinflammation (Diaz-Casado *et al*, 2018). The embryos were exposed to MPTP from 24 hpf to 72 hpf, and β-RA was added to the wells from 72 hpf to 120 hpf. After completing the experiment protocol, at 120 hpf, the mortality rate was calculated as the percentage of dead embryos with respect to total embryos. Edema, the most common malformations, as well as tail and yolk abnormalities, were macroscopically quantified (Diaz-Casado *et al*, 2018). Adult zebrafish (*Danio rerio*) of the AB line were provided by ZFBiolabs S.L (Madrid, Spain) and used as breeding stocks. The fish line was maintained in the University of Granada's facility at a water temperature of 28.5 ± 1°C and under a photoperiod of 14:10 h (lights on at 08:00 h) in a recirculation aquaculture system (Aquaneering Incorporated, Barcelona, Spain).

Experiments in mice were performed according to a protocol approved by the Institutional Animal Care and Use Committee of

the University of Granada (procedures 9-CEEA-OH-2013) and were in accordance with the European Convention for the Protection of Vertebrate Animals used for Experimental and Other Scientific Purposes (CETS # 123) and the Spanish law (R.D. 53/2013). Experiments in zebrafish were performed according to a protocol approved by the Institutional Animal Care and Use Committee of the University of Granada (procedures CEEA 2010-275).

### Histology and immunohistochemistry

Multiple brain sections (4 μm thickness) were deparaffinized with xylene and stained with hematoxylin and eosin (H&E). Immunohistochemistry was carried out in the same sections, using the following primary antibodies: anti-GFAP (Millipore, MAB360, dilution 1:400) and anti-Iba-1 (Wako, 019-19741, dilution 1:500). The detection of carbonyl groups as a marker of protein oxidation was performed using a commercial kit (OxyIHC™ Oxidative Stress Detection Kit; S7450, Millipore; Garcia-Corzo et al, 2013).

Type I and type II skeletal muscle fibers were examined through histochemical detection of cytochrome oxidase (COX) and succinate dehydrogenase (SDH) enzymes activity (Tanji & Bonilla, 2008).

### In vivo MRI and proton MRS

Magnetic resonance imaging and MRS studies were conducted using a 7 T horizontal bore magnet Bruker Biospec TM 70/20 USR designed for small animal experimentation. Following localizer scans, high-resolution axial and coronal T2-weighted datasets were acquired in order to visualize any cerebral or structural atrophy and to investigate for potential focal pathologies (Diaz et al, 2012).

### Transcriptome analysis by RNA-Seq

The RNeasy Lipid Tissue Mini Kit (Qiagen) was used to extract total RNAs from the brainstem of five animals in each experimental group. The RNAs were precipitated and their quality and quantity assessed using an Agilent Bioanalyzer 2100 and an RNA 6000 chip (Agilent Technologies). The cDNA libraries were then constructed using the TruSeq RNA Sample Prep Kit v2 (Illumina, Inc.) and their quality checked using an Agilent Bioanalyzer 2100 and a DNA 1000 chip (Agilent Technologies). The libraries were Paired End sequenced in a HiSeq 4000 system (Illumina, Inc.). We aimed for 4–5 Giga Bases outcome per sample. The quality of the resulting sequencing reads was assessed using FastQC. The GRCm38.p5 fasta and gtf files of the reference mouse genome were downloaded from the Ensembl database and indexed using the bwtsw option of BWA (Linden et al, 2012). BWA, combined with xa2multi.pl and SAMtools (Li et al, 2009), was also used for aligning the sequencing reads against the reference genome, and HTSeq was used for counting the number of reads aligned to each genomic locus (Mishanina et al, 2015). The alignments and counting were carried out in our local server following the protocols as described (Brzywczy et al, 2002; Di Meo et al, 2011).

After elimination of the genomic loci that aligned to < 5 reads in < 5 samples and normalization of the read counts by library size, the differential gene expression was detected using the Generalized Linear Model (glmLRT option) statistic in EdgeR (Hine et al, 2015). We used a 0.05 P-level threshold after False Discovery Rate

correction for type I error. The heatmap figure was made using the heatmap function in R (https://www.r-project.org/). Annotation of the differentially expressed genes was obtained from the Mouse Genome Informatics (http://www.informatics.jax.org/).

### Mitochondrial respiration

To isolate fresh mitochondria, mice were sacrificed and the organs were immediately extracted and placed in ice. The final crude mitochondrial pellet was re-suspended in MAS 1X medium (Luna-Sanchez et al, 2015). Mitochondrial respiration was measured by using an XFe24 Extracellular Flux Analyzer (Seahorse Bioscience; Rogers et al, 2011). Respiration by the mitochondria was sequentially measured in a coupled state with the substrates (succinate, malate, glutamate, and pyruvate) present (basal respiration or State 2), followed by State 3o (phosphorylating respiration, in the presence of ADP and substrates); State 4 (non-phosphorylating or resting respiration) was measured after addition of oligomycin when all ADP was consumed, and then maximal uncoupler-stimulated respiration (State 3u; Luna-Sanchez et al, 2015). All data were expressed in pmol/min/mg protein.

### CoQ-dependent respiratory chain activities

Coenzyme Q-dependent respiratory chain activities were measured in submitochondrial particles, as previously described (Garcia-Corzo et al, 2014). To test whether β-RA works as a CoQ substitute en the mitochondrial respiratory chain, we also measure the CI-III activity after adding 50 μM β-RA into the reaction mix. The results were expressed in nmol reduced cyt c/min/mg prot.

### Evaluation of supercomplexes formation by BNGE

Blue native gel electrophoresis (BNGE) was performed on crude mitochondrial fractions from mice kidneys and brain. Mitochondria were isolated and analyzed by BNGE, as previously described (Fernandez-Vizarra et al, 2002). After electrophoresis, the complexes were electroblotted onto PVDF membranes and sequentially tested with specific antibodies against CI, anti-NDUFA9 (Abcam, ab14713, dilution 1:5,000), CIII, anti-ubiquinol-cytochrome c reductase core protein I (Abcam, ab110252, dilution 1:5,000), and VDAC1 (Abcam, ab14734, dilution 1:5,000).

### Sample preparation and Western blot analysis in tissues

For Western blot analyses in mouse tissues, samples were homogenized and sonicated in T-PER® buffer (Thermo Scientific) with protease inhibitor cocktail (Pierce). For Western blot analyses in brain mitochondria, the pellets containing the mitochondrial fraction were re-suspended in RIPA buffer with protease inhibitor cocktail. Protein band intensity was normalized to VDAC1 (mitochondrial proteins), and the data were expressed in terms of percent relative to wild-type mice or control cells (Luna-Sanchez et al, 2015, 2017). The following primary antibodies were used: anti-SQRDL (Proteintech, 17256-1-AP, dilution 1:500), anti-PDSS2 (Proteintech, 13544-1-AP, dilution 1:500), anti-COQ2 (Origene, TA341982, dilution 1:2,500), anti-COQ4 (Proteintech, 16654-1-AP, dilution 1:2,000), anti-COQ5

## The paper explained

### Problem

$CoQ_{10}$ deficiency is a mitochondrial syndrome with heterogeneous clinical presentations. Typically, the patients with neurological symptoms or the cases with multisystemic affection respond poorly to the high doses of exogenous $CoQ_{10}$ supplementation due to the limitation of this molecule to cross the blood–brain barrier. Therefore, an alternative therapeutic option is needed for these patients.

### Results

In this study, we have used a mouse model of mitochondrial encephalopathy due to CoQ deficiency to test a novel therapy based in the administration of β-resorcylic acid (β-RA), an analog of 4-hydroxybenzoic acid (4-HB), the precursor of the endogenous CoQ biosynthesis. In peripheral tissues, the β-RA was able to reduce the levels of demethoxyubiquinone, a CoQ biosynthetic intermediate metabolite that is toxic for the mitochondrial respiratory chain. Consequently, the mitochondrial bioenergetics performance was improved in peripheral tissues during β-RA treatment, leading to a reduction in the spongiosis and astrogliosis in the brain. As a result, the mutant mice treated with β-RA reached a maximum lifespan of 25 months of age, with a median survival of 22 months of age. This represents a significant increase in the animal survival if we compare the data to those of the untreated mutant mice, which reached a maximum age of 7 months, with a median survival of 5 months of age; or with the $CoQ_{10}$-treated (in its reduced form, ubiquinol-10) mutant mice, which reached a maximum age of 17 months, with a median survival of 13 months of age.

### Impact

The results support the use of β-RA at least as a compassionate option in cases of severe CoQ deficiencies due to mutations in *COQ9* or *COQ7*. Therefore, these findings deserve additional in depth research in order to translate the use of β-RA and/or other 4-HB analogs into the clinical practice.

(Proteintech, 17453-1-AP, dilution 1:1,000), anti-COQ6 (Proteintech, 12481-1-AP, dilution 1:500), anti-COQ7 (Proteintech, 15083-1-AP, dilution 1:1,000), anti-COQ8A (Proteintech, 15528-1-AP, dilution 1:2,500), anti-S6R (Cell Signaling, 2217, dilution 1:1,000), anti-S6RP (Cell Signaling, 2211, dilution 1:1,000), anti-FGF21 (Abcam, ab171941, dilution 1:1,000), anti-VDAC1 (Abcam, ab14734, dilution 1:5,000), anti-TOM20 (Proteintech, 11802-1-AP, dilution 1:5,000), and anti-GAPDH (Santacruz, sc-166574, dilution 1:200).

### Quantification of $CoQ_9$ and $CoQ_{10}$ levels in mice tissues

$CoQ_9$ and $CoQ_{10}$ levels were determined via reversed-phase HPLC coupled to electrochemical (EC) detection (Lopez *et al*, 2010; Garcia-Corzo *et al*, 2013). The results were expressed in ng CoQ/mg prot.

### Quantification of β-RA levels and screening of potential derivate molecules in mice tissues

The levels of β-RA were quantified tissue extracts by HPLC-UV. The screening to detect other hydroxybenzoic acid derivate was performed by UPLC-MS/MS.

### Quantification of pro-inflammatory mediators in murine RAW 264.7 macrophages and brainstem extracts

Concentrations of interleukin (IL)-1β, IL-2, interferon-γ (IFN-γ), tumor necrosis factor-α (TNF-α), and growth regulated oncogene-α (GRO-α) were quantified in duplicates in the supernatant of the cell culture of murine RAW 264.7 macrophages and brainstem tissue using ProcartaPlexTM Multiplex immunoassays (eBioscience), as previously described and according to the manufacturer instructions (Hinz *et al*, 2000; Reichmann *et al*, 2015).

### Statistical analysis

Number of animals in each group were calculated in order to detect gross ~60% changes in the biomarkers measurements (based upon $\alpha = 0.05$ and power of $\beta = 0.8$). We used the application available in: http://www.biomath.info/power/index.htm. Animals were genotyped and randomly assigned in experimental groups in separate cages by the technician of the animal facility. Data are expressed as the mean ± SD of 6–10 experiments per group. A one-way ANOVA with a Tukey's *post hoc* test was used to compare the differences between three experimental groups. Studies with two experimental groups were evaluated using unpaired Student's *t*-test. A *P*-value of < 0.05 was considered to be statistically significant. A Mann–Whitney *U*-test was used for the statistic analysis of the hormones measurements. Survival curve was analyzed by log-rank (Mantel-Cox) and the Gehan–Breslow–Wilcoxon tests.

## Data and software availability

RNA-Seq data were generated as described above. The files have been uploaded in the repository Gene Expression Omnibus. The accession number is GSE120287. All data can be found at https://www.ncbi.nlm.nih.gov/geo/query/acc.cgi?acc = GSE120287.

Expanded View for this article is available online.

## Acknowledgements

We thank Stacy Kelly Aguirre for the English editing. We are grateful to Ana Fernandez (Universidad de Granada) and Samuel Cantarero (Universidad de Granada) for their technical support at the facilities of bioanalysis, MRI and MS. We thank Dr. Catarina M. Quinzii for the critical review of the study. This work was supported by grants from Ministerio de Economía y Competitividad, Spain, and the ERDF (grant numbers SAF2013-47761-R and SAF2015-65786-R), from the NIH (P01HD080642), and from the University of Granada (grant reference "UNETE", UCE-PP2017-06). A.H.-G. is a "FPU fellow" from the Ministerio de Educación Cultura y Deporte, Spain. E.B.-C- and M.E.D.-C. were supported by the Junta de Andalucía. L.C.L. was supported by the "Ramón y Cajal" National Programme, Ministerio de Economía y Competitividad, Spain (RYC-2011-07643). The results shown in this article will constitute a section of the A.H.-G. doctoral thesis at the University of Granada.

## Author contributions

AH-G led the study, developed the survival assay and the body weight measurement, and conducted WB assays, the tests to assess the mitochondrial bioenergetics, HPLC analysis, luminex assays, immunohistochemistry assays, analyzed the results, designed the figures, and wrote the manuscript.

EB-C collected the data for the survival curve and the body weight representation, performed the work in the RAW cells, contributed to the mitochondrial assays, managed the mouse colony, and critically reviewed the manuscript. MB designed and analyzed the RNA-Seq and contributed to the discussion. MED-C developed the assays with zebrafish embryos and critically reviewed the manuscript. LS-M ran some WB in the kidneys and heart. MR and JD measured the blood pressure and heart rate. RKS performed the COX and SDH assays in muscle. CP measured the levels of the steroid hormones. GE and DA-C contributed to the discussion. LCL conceived the idea for the project, supervised the experiments, and edited the manuscript. All authors critically reviewed the manuscript. The results shown in this article will constitute a section of the AH-G doctoral thesis at the University of Granada.

## Conflict of interest

A.H.-G., E.B.-C., M.B., and L.C.L. are inventors on a patent application entitled "Pharmacological Composition for the Treatment of Diseases Associated to a Deficiency in the Endogenous Cellular Synthesis of Coenzyme Q9 and Coenzyme Q10".

## For more information

(i)    Online Mendelian Inheritance in Man (OMIM) COQ9: *S. cerevisiae*, homolog of; COQ9: http://omim.org/entry/612837#0001

(ii)   International mito-patients: http://www.mitopatients.org/index.html

(iii)  The association of mitochondrial disease patients in Spain: http://www.aepmi.org/publicoIngles/index.php

(iv)   The United Mitochondrial Disease Foundation: http://www.umdf.org/

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
