## [Review Process File · EMBO Molecular Medicine]

β -RA reduces DMQ/CoQ ratio and rescues the encephalopathic phenotype in *Coq9*^{R239X} mice

Agustín Hidalgo-Gutiérrez, Eliana Barriocanal-Casado, Mohammed Bakkali, M. Elena Díaz-Casado, Laura Sánchez-Maldonado, Miguel Romero, Ramy K. Sayed, Cornelia Prehn, Germaine Escames, Juan Duarte, Darío Acuña-Castroviejo, Luis C. López

Review timeline:

Submission date:	23 June 2018
Editorial Decision:	26 July 2018
Revision received:	27 September 2018
Editorial Decision:	19 October 2018
Revision received:	23 October 2018
Accepted:	26 October 2018

Editor: Céline Carret

Transaction Report:

1st Editorial Decision

26 July 2018

Thank you for the submission of your manuscript to EMBO Molecular Medicine. We have now heard back from the three referees whom we asked to evaluate your manuscript.

You will see from the set of reports pasted below that the three referees find the study to be of interest and are overall supportive of publication. However, details, clarifications and further discussions are needed. In addition, different dosing and duration of treatment should be investigated (ref.2 and 3), which we agree would increase the translational potentials of the findings. Also, both ref.2 and 3 suggest modifying the title and abstract and ref.3 highlights a few overstatements that should be tuned down. Finally, upon our cross-commenting exercise, a metabolic profile analysis was suggested as well as showing data on NDUFS4 KO mice w/o treatment (as ref.3 requested).

We would welcome the submission of a revised version within three months for further consideration and would like to encourage you to address all the criticisms raised as suggested to improve conclusiveness and clarity. Please note that EMBO Molecular Medicine strongly supports a single round of revision and that, as acceptance or rejection of the manuscript will depend on another round of review, your responses should be as complete as possible.

I look forward to receiving your revised manuscript.

***** Reviewer's comments *****

Referee #1 (Remarks for Author):

This study concerns a mouse model of primary Coenzyme Q deficiency, where the synthesis of CoQ is impaired by lack of the hydroxylation step that is required for the formation of the 2-methoxy group. In the mouse model as well as in the human disease induced by genetic defect of COQ9 (a protein associated to the hydroxylase COQ7), the synthesis of the final CoQ molecules (CoQ9 in mouse, CoQ10 in humans) is impaired, with accumulation of 2-demethoxy CoQ (DMQ). The affected animals are CoQ-deficient and develop a fatal encephalopathy. Since previous attempts to cure the disease with exogenous ubiquinone or ubiquinol were scarcely successful, the authors have developed a strategy to circumvent the lacking step by using as a precursor the 2-hydroxyl form of 4-hydroxybenzoate, i.e. β -resorcylic acid (RA) that has an hydroxyl residue in the 2-position. RA was very effective in rescuing the encephalopathy, leading to significant increase of survival.

To understand the mechanism of this effect, the authors have accomplished an extensive study of bioenergetics and CoQ levels in brain and other tissues (kidney, heart, muscle, liver) in wild type and mutant mice and in mutant mice supplemented with RA. In brain, the CoQ levels and respiratory activities were strongly lowered, while DMQ was strongly accumulated, but RA had no effect whatsoever, despite the striking effect in curing the encephalopathy. The effects on brain were at difference with other tissues studied, and mainly in kidney: in all these tissues, the levels of DMQ decreased, but the levels of CoQ9 and CoQ10 were raised very little (kidney) or not at all. The activities of NADH cytochrome c reductase (I + III) and of succinate cytochrome c reductase (II + III) were decreased in all tissues, but were partially or totally recovered after RA supplementation. The levels of the CoQ biosynthetic enzymes forming a complex were decreased in the mutants and were restored by RA, suggesting that it contributes to stabilize the complex.

The authors conclude that the effects of mutation are largely affected through accumulation of DMQ, and RA mainly acts by lowering DMQ levels. The beneficial effect on the encephalopathy, however, must have a different origin and may be due to interactions between organs, exerting an anti-inflammatory effect.

The manuscript is very analytical and the experiments are scrupulously performed with appropriate controls. The main message, that DMQ levels are responsible for the pathological effects, is plausible, although the reasons given to explain the major findings are not always convincing. The authors should reply to the following points.

- a. Treatment with RA is able to lower or to cancel the levels of DMQ without any significant increase of the synthesis of CoQ9. The explanation given of a possible decrease (increase?) of Km of RA is not tenable. The reference to Pierrel is not correct, since in the paper quoted he finds an increase of CoQ9.
- b. Despite the low level of CoQ maintained after RA treatment, the activities of NADH cytochrome c reductase and succinate cytochrome c reductase are restored in many tissues. The authors correctly attribute this finding to the lowering of DMQ and suggest that DMQ is an inhibitor of Complex I: indeed the cited paper by Yang et al shows inhibition of NADH cyt.c reductase by DMQ. However, the same authors find no inhibition of succinate cyt.c reductase by DMQ, whereas in this study both activities are depressed in mutant mice and restored in RA-treated mice. How can this be explained?
- c. Methods: mitochondrial respiration. What is the substrate used?
- d. Finally there are some errors of English throughout the manuscript.

Referee #2 (Comments on Novelty/Model System for Author):

The animal model used for this study is appropriate, although more details should have been provided to explain the choice of doses of Resorcylic Acid (β -RA) used in this study and the duration of treatment. Furthermore, there is no information on whether a higher dose of Resorcylic Acid or a longer term treatment may induce further biochemical/histological improvement in the animal model.

Referee #2 (Remarks for Author):

This is a very interesting paper which describes the efficacy of Resorcylic Acid (β -RA) in the treatment of the mouse model of CoQ9 deficiency.

I have some comments for the authors to address:

1. The actual title of the paper doesn't fully convey the study and an alternative title may be preferable.
2. The abstract is a little vague and doesn't contain any actual numerical results and should be amended.
3. In the study, why was the dosage and treatment time of Resorcylic Acid (β -RA) selected? Did the authors try higher doses or longer treatment times to try to induce further improvements?
4. Is Resorcylic Acid able to act as an electron carrier in the respiratory chain or function as an antioxidant? If so, this may also help to explain some of the findings of the study.
5. In view of the possible inhibitory effect of demethoxyubiquinone 9 or 10 on respiratory chain function, did the authors investigate this possibility? I realise a reference is cited to justify the possible inhibitory effect on the respiratory chain but were the levels of demethoxyubiquinone 9 in the animal model high enough to cause such inhibition?
5. Can the authors be sure that Resorcylic Acid itself and not a metabolite of this molecule is having the therapeutic effect?
6. Does Resorcylic Acid have any inhibitory effect on complex I-III or II-III that may be influencing the enzyme assay results?
7. In view of the possibility that some possible peripheral effect is inducing some cerebral improvement in the animals following Resorcylic Acid treatment, did the authors examine the effect of Resorcylic Acid on the blood brain barrier or whether it could improve ubiquinol/CoQ10 uptake into the brain?

Referee #3 (Remarks for Author):

Hidalgo-Gutiérrez et al. describes the various effects of beta-RA on a Coq9R239X mouse, which is a model mouse of human Q deficiency. A Coq9R239X mouse does not survive longer than 7 months, but they showed a significant extension of the survival of Coq9R239X mouse by administration of beta-RA which is better than Q10 administration. The effect of beta-RA on the survival on a Coq9R239X mouse is striking. To understand the underlying mechanisms of this effect, they analyzed in various ways, histopathologically, biochemically, transcriptomically and physiologically, which were very well conducted. They did not see the increase of Q9 by administration of beta-RA, which is different from the previous findings in a Coq7 knockout mouse done by Wang et al. 2015 and Freyer et al. 2015. It is an intriguing difference. They concluded beta-RA affected the ratio of DMQ/CoQ and stability of CoQ biosynthetic proteins. This is a very nice work that I can basically recommend for publication.

1) I suggest the authors reconsidering the title 'The importance of DMQ/CoQ Ratio and Complex Q Stability in the Treatment of CoQ Deficiency' This is a little broad to speculate what was found in this work. There are no words of 'betaRA' and 'mouse Coq9R239X'. The Figure 6 data describing the increased level of Coq proteins by beta-RA are not convincing to me. It has some effects on specific proteins but is not observed in the brain. I think it is a little earlier to conclude by putting 'stability' in the title.

2) This work focused on Coq9R239X mice, yet it is not clearly stated in the abstract. The Coq9R239X mice should be included in the sentence of the abstract.

3) Did they test any dose dependent effects of beta-RA or just tested by 1%? Is there any reason for choosing this concentration?

4) Page5, line 18: A more recent review such as 'Kawamukai, Biosci Biotechnol Biochem. 2016;80(1):23-33' is better to be cited.

5) Page 10, line 20: do >> does

6) Page 11, line 19: Fig S5 >> Fig S6

- 7) Page 15, line13: levels androstenodione >> levels of androstenodione
- 8) Page 16, line 1:' a novel treatment for primary and secondary'.
--- This study only shows the effect of RA on a primary CoQ deficiency, not secondary CoQ deficiencies.
- 9) Page 20, line 17-22: There are unnecessary italicized sentences.
- 10) Page 28, 'The study provides a powerful therapy for patients with mutations in COQ9, COQ7 or COQ4'
---- I think this is an over statement. The results of COQ7 mice were previously shown by Wang et al. 2015 and Freyer et al. 2015. I do not see the data related on COQ4 mutants in this work and previous works.
- 11) Page 28, 'This work demonstrates the specific therapeutic mechanism of β -RA in the Coq9R239X mouse model, opening the application of 4-HB analogs to secondary CoQ deficiencies due to decreased levels of CoQ biosynthetic proteins'
--- I think this is also an overstatement. I do not see any results supporting therapeutic effect on secondary CoQ deficiencies. The effect of beta-RA on stability of CoQ biosynthetic proteins is still arguable. The level of CoQ7 protein does not change in any tested organs in Figure 6.
- 11) Page 34, line 11: 'Smith AC --- ' volume and page number?
- 12) Figure 1G: Is this result based on both female and male or either? It is better to clarify.
- 14) Figure 3I: Is there data of Ndufs4 knockout mice of untreated?
- 15) Figure 5: Coq8 >> Coq8A
- 16) Fig. S1: The isoprenoid part needs to be amended. The left parenthesis should be moved to the left. Isoprene has 5 carbons. The nomenclature of 5HQ (hydroxyquinone) is not correct.
- 17) Fig S11 F: I see that mFGF21 is clearly higher in beta-RA treated mice than untreated. What would be the quantification if the bands of mFGF21 (but not preFGF21) were compared?

1st Revision - authors' response

27 September 2018

Referee #1

We thank the general positive comments of this reviewer, with a special note in the quality of the analysis and the experiments. We also appreciate the useful recommendations to improve the manuscript.

Treatment with RA is able to lower or to cancel the levels of DMQ without any significant increase of the synthesis of CoQ9. The explanation given of a possible decrease (increase?) of Km of RA is not tenable. The reference to Pierrel is not correct, since in the paper quoted he finds an increase of CoQ9.

We thank the reviewer for noting the inaccurate interpretation of our results. We think that a high concentration of b-RA allows this molecule to compete with 4-HB in order to enter in the reaction catalyzed by COQ2. However, the km of b-RA in this reaction should be higher (not lower) than the km of 4-HB, the natural compound for this reaction. In this context, b-RA should be substantially more abundant than 4-HB in order to produce more CoQ, as it has been suggested by Pierrel (2017), and as we can now confirm in our metabolic screening. Consequently, we believe that b-RA reaches enough levels in kidneys to increase the CoQ biosynthesis. In other peripheral tissues, however, the levels of b-RA are not enough to produce more CoQ. Nevertheless, because of the higher km of b-RA, the levels of DMQ are decreased not only in kidneys but also in heart, muscle and liver. These data are now confirmed with the analyses using the double dose of b-RA (Table EV1).

Despite the low level of CoQ maintained after RA treatment, the activities of NADH cytochrome c reductase and succinate cytochrome c reductase are restored in many tissues. The authors correctly attribute this finding to the lowering of DMQ and suggest that DMQ is an inhibitor of Complex I: indeed the cited paper by Yang et al shows inhibition of NADH cyt.c reductase by DMQ. However, the same authors find no inhibition of succinate cyt.c reductase by DMQ, whereas in this study both activities are depressed in mutant mice and restored in RA-treated mice. How can this be explained?

We thank the reviewer for cleverly pointing out those differences. The article published by Yang and colleagues demonstrates that DMQ inhibits the activity of one of the two CoQ-dependent complexes activities (CI+III) in worms. This may reflect the competition of DMQ and CoQ for the same binding sites, together with the inability of DMQ to transfer electrons, as it has been reported by different groups (Arroyo et al, 2006). Our results suggest that DMQ inhibits the two CoQ-dependent complexes activities (CI+III and CII+III) in mice. We believe that those differences could be attributed to the different structure and functionality of the mitochondrial respiratory chain in different organisms (worms vs. mice), tissues and cell types (Vafai & Mootha, 2012). We have added a couple of sentences in the discussion to clarify this point (p. 18-19).

Methods: mitochondrial respiration. What is the substrate used?

The detailed description of the mitochondrial respiration measurement was reported in a previous article (Luna-Sanchez et al, 2015), and it is included in the appendix supplemental material and methods. We have used a combination of substrates (succinate, malate, glutamate and pyruvate) and we have added this information in the main methods section (p. 25).

Finally there are some errors of English throughout the manuscript.

A native English speaker has reviewed the manuscript language.

Referee #2

We would like to thank the reviewer for the positive comments and for sharing the enthusiasm and interest in this study. We also appreciate the insightful comments of this reviewer.

The actual title of the paper doesn't fully convey the study and an alternative title may be preferable.

We agree with the reviewer and the title has been changed to "b-RA reduces DMQ/CoQ ratio and rescues the encephalopathic phenotype in *Coq9^{R239X}* mice".

The abstract is a little vague and doesn't contain any actual numerical results and should be amended.

We understand the reviewer's point of view. However, EMBO Molecular Medicine limits the abstract to a maximum of 175 words, making difficult to include a more detailed explanation of the results. Nevertheless, to partially address the reviewer's comment, we have added the name of the therapeutic molecule and the mouse model in the title and in the abstract.

In the study, why was the dosage and treatment time of Resorcylic Acid (β -RA) selected? Did the authors try higher doses or longer treatment times to try to induce further improvements?

We have clarified the reason for choosing the dose in the experimental design section (p. 22). We chose the dose of β -RA based on the results of the treatment of the *Coq7* conditional KO model. In that study, 1 g β -RA/kg b.w./day was enough to increase the levels of CoQ and the survival of *Coq7* KO mice. We have not tried any other dose but this is something that should be done in the future. Nevertheless, we have done a pilot study to measure the levels of CoQ and DMQ after one month of treatment with the double dose of b-RA (2 g/kg b.w./day), as well as with the regular dose after 9 months of treatment (Table EV1; Figures S6 and S7). The most significant finding in the three situations is the reduction in the levels of DMQ₉ and, consequently, in the DMQ₉/CoQ₉ ratio. However, no major differences were found between the three experimental conditions (p. 13).

Is Resorcylic Acid able to act as an electron carrier in the respiratory chain or function as an antioxidant? If so, this may also help to explain some of the findings of the study.

We have measured the CI+III activity after adding 50 mM b-RA in the reaction mix. Our results show that b-RA cannot act as an electron carrier in the mitochondrial respiratory chain (p. 11; Appendix Fig S4).

In view of the possible inhibitory effect of demethoxyubiquinone 9 or 10 on respiratory chain function, did the authors investigate this possibility? I realize a reference is cited to justify the possible inhibitory effect on the respiratory chain but were the levels of demethoxyubiquinone 9 in the animals model higher enough to cause such inhibition?

We also believe that the effect of DMQ in the mitochondrial respiratory chain should be further investigated. Unfortunately, DMQ is not commercially available and that creates a limitation for additional mechanistic studies.

Can the authors be sure that Resorcylic Acid itself and not a metabolite of this molecule is having the therapeutic effect?

We have performed a metabolic screening in order to identify other hydroxybenzoic acids present in the samples. We were only able to detect the natural 4-HB (Appendix Table S1); and no statistic differences were found between the experimental groups. Therefore, we believe that the therapeutic effect is mediated by b-RA.

Does Resorcylic Acid have any inhibitory effect on complex I-III or II-III that may be influencing the enzyme assay results?

We have measured the CI+III activity after adding 50 mM b-RA in the reaction mix. Our results show that b-RA cannot act as an electron carrier in the mitochondrial respiratory chain (p.13-14; Appendix Fig S4), probably because, among other things, it does not have the polyprenoid tail characteristic of CoQ₉ or CoQ₁₀.

In view of the possibility that some possible peripheral effect is inducing some cerebral improvement in the animals following Resorcylic Acid treatment, did the authors examine the effect of Resorcylic Acid on the blood brain barrier or whether it could improve ubiquinol/CoQ10 uptake into the brain?

These are two interesting hypothesis that could be tested in the future. Nevertheless, we have not found any data in the literature suggesting that b-RA (or other analog) may have some impact in the BBB. Also it is very unlikely that b-RA could improve CoQ₁₀ uptake into the brain because the levels of CoQ₉ and CoQ₁₀ did not change in the brain during the treatment with b-RA.

Referee #3

We thank the general positive comments of this reviewer, as well as the very useful recommendations to improve the manuscript.

I suggest the authors reconsidering the title 'The importance of DMQ/CoQ Ratio and Complex Q Stability in the Treatment of CoQ Deficiency' This is a little broad to speculate what was found in this work. There are no words of 'betaRA' and 'mouse Coq9R239X'. The Figure 6 data describing the increased level of Coq proteins by beta-RA are not convincing to me. It has some effects on specific proteins but is not observed in the brain. I think it is a little earlier to conclude by putting 'stability' in the title.

We agree with the reviewer and we now propose the following title: “b-RA reduces DMQ/CoQ ratio and recues the encephalopathic phenotype in Coq9^{R239X} mice”.
This work focused on Coq9R239X mice, yet it is not clearly stated in the abstract. The Coq9R239X mice should be included in the sentence of the abstract.

We have included the term “Coq9^{R239X}” in the abstract.

Did they test any dose dependent effects of beta-RA or just tested by 1%? Is there any reason for choosing this concentration?

We have clarified the reason of choosing the dose in the experimental design section (p. 22). We chose the dose of β -RA based on the results of the treatment of the *Coq7* conditional KO model. In that study, 1 g β -RA/kg b.w./day was enough to increase the levels of CoQ and the survival of *Coq7* KO mice. We have not tried any other dose but this is something that should be done in the future. Nevertheless, we have done a pilot study to measure the levels of CoQ and DMQ after one month of treatment with the double dose of β -RA (2 g/kg b.w./day), as well as with the regular dose after 9 months of treatment (Table EV1; Figures S6 and S7). The most significant finding in the three situations is the reduction in the levels of DMQ₉ and, consequently, in the DMQ₉/CoQ₉ ratio. However, no major differences were found between the three experimental conditions (p. 13).

Page 5, line 18: A more recent review such as 'Kawamukai, Biosci Biotechnol Biochem. 2016;80(1):23-33' is better to be cited.

We have updated the references with this more recent review about CoQ biosynthesis.

Page 10, line 20: do >> does

We have corrected this error.

Page 11, line 19: Fig S5 >> Fig S6

We have corrected this error.

Page 15, line13: levels androstenodione >> levels of androstenodione

We have corrected this error.

Page 16, line 1: 'a novel treatment for primary and secondary'. --- This study only shows the effect of RA on a primary CoQ deficiency, not secondary CoQ deficiencies.

We agree with the reviewer, so we have removed the term “secondary”.

Page 20, line 17-22: There are unnecessary italicized sentences.

We have corrected this error.

Page 28, 'The study provides a powerful therapy for patients with mutations in COQ9, COQ7 or COQ4' ---- I think this is an over statement. The results of COQ7 mice were previously shown by Wang et al. 2015 and Freyer et al. 2015. I do not see the data related on COQ4 mutants in this work and previous works.

Even if DMQ has been recently detected in samples from patients with *COQ4* mutations, we agree with the cautious perspective of the reviewer, so we have removed the reference to a potential application to patients with mutations in *COQ4*.

Page 28, 'This work demonstrates the specific therapeutic mechanism of β -RA in the Coq9R239X mouse model, opening the application of 4-HB analogs to secondary CoQ deficiencies due to decreased levels of CoQ biosynthetic proteins' --- I think this is also an overstatement. I do not see any results supporting therapeutic effect on secondary CoQ deficiencies. The effect of beta-RA on stability of CoQ biosynthetic proteins is still arguable. The level of CoQ7 protein does not change in any tested organs in Figure 6.

We agree with the reviewer, so we have reformulated the sentence as follow: “This work demonstrates the specific therapeutic mechanism of β -RA in the *Coq9^{R239X}* mouse model, opening the potential application of 4-HB analogs to other forms of CoQ deficiencies”.

Page 34, line 11: 'Smith AC --- ' volume and page number?

We have added the volume and page numbers to this reference.

Figure 1G: Is this result based on both female and male or either? It is better to clarify.

Yes, the data are from both male and female mice. We have added the following sentence: "Data from male and female mice are represented together."

Figure 3I: Is there data of Ndufs4 knockout mice of untreated?

We have now included the survival curve of the untreated *Ndufs4*^{-/-} mice. As expected, we did not find major differences between untreated and treated mice.

Figure 5: Coq8 >> Coq8A

We have corrected this typo.

Fig. S1: The isoprenoid part needs to be amended. The left parenthesis should be moved to the left. Isoprene has 5 carbons. The nomenclature of 5HQ (hidroxyquinone) is not correct.

We apologize for the errors in this figure. We have made the appropriated corrections.

Fig S11 F: I see that mFGF21 is clearly higher in beta-RA treated mice than untreated. What would be the quantification if the bands of mFGF21 (but not preFGF21) were compared?

It is true that there is a trend toward increased levels of mFGF21 in the liver, as it is shown in the quantification. However, there are no statistic differences due to the high variability between samples of the same experimental groups.

We hope that the editor and the reviewers find our responses satisfactory.

References

Arroyo A, Santos-Ocana C, Ruiz-Ferrer M, Padilla S, Gavilan A, Rodriguez-Aguilera JC, Navas P (2006) Coenzyme Q is irreplaceable by demethoxy-coenzyme Q in plasma membrane of *Caenorhabditis elegans*. *FEBS Lett* **580**: 1740-1746

Luna-Sanchez M, Diaz-Casado E, Barca E, Tejada MA, Montilla-Garcia A, Cobos EJ, Escames G, Acuna-Castroviejo D, Quinzii CM, Lopez LC (2015) The clinical heterogeneity of coenzyme Q10 deficiency results from genotypic differences in the *Coq9* gene. *EMBO Mol Med* **7**: 670-687

Vafai SB, Mootha VK (2012) Mitochondrial disorders as windows into an ancient organelle. *Nature* **491**: 374-383

2nd Editorial Decision

19 October 2018

Thank you for the submission of your revised manuscript to EMBO Molecular Medicine. We have now received the enclosed reports from the referees that were asked to re-assess it. As you will see the reviewers are now globally supportive and I am pleased to inform you that we will be able to accept your manuscript pending the following minor editorial amendments.

***** Reviewer's comments *****

Referee #1 (Comments on Novelty/Model System for Author):

This is a very good study, that provides a novel approach and has important medical outcomes.

Referee #1 (Remarks for Author):

The author has answered to all the queries of this reviewer in an exhaustive and convincing manner.

Referee #2 (Comments on Novelty/Model System for Author):

Please refer to my previous comments on this manuscript.

Referee #2 (Remarks for Author):

The authors have addressed all my comments appropriately and have amended the manuscript accordingly including the information I requested. I therefore feel that the paper is now appropriate for publication.

Referee #3 (Remarks for Author):

The responses to my previous comments are generally acceptable.

The structures of Decaprenyl diphosphate and the side chains of prenylated PHB, DMQ, DMeQ, and CoQ in Figure EV1 are not correct. Need amendment.

Corresponding Author Name: Luis C. López
Journal Submitted to: EMBO Molecular Medicine
Manuscript Number: EMM-2018-09466